# New Directions in the Therapy of Glioblastoma

**DOI:** 10.3390/cancers14215377

**Published:** 2022-10-31

**Authors:** Katarzyna Szklener, Marek Mazurek, Małgorzata Wieteska, Monika Wacławska, Mateusz Bilski, Sławomir Mańdziuk

**Affiliations:** 1Department of Clinical Oncology and Chemotherapy, Medical University of Lublin, 8 Jaczewski Street, 20-090 Lublin, Poland; 2Department of Neurosurgery, Medical University of Lublin, 20-090 Lublin, Poland; 3Department of Radiotherapy, Medical University of Lublin, 20-090 Lublin, Poland

**Keywords:** glioblastoma, brain tumor, targeted therapies, molecular oncology

## Abstract

**Simple Summary:**

Despite the continuous development of treatment technology, glioblastoma remains a challenge for modern medicine. The standard of care remains a radical surgical resection combined with adjuvant radiotherapy and chemotherapy with temozolomide. However, the usage of these procedures is not associated with satisfactory treatment responses. Therefore, it is important to develop systemic therapies that support treatment and sometimes represent the only therapeutic option. Molecular targeted therapy and immunotherapy appear to be encouraging approaches. In recent years, a number of key metabolic points for the progression of glioblastoma have been identified. In this review, we presented the therapeutic targets that are currently under study for new clinical trials. The aim of our study was to present and analyze the up-to date literature describing the epidemiology, genetics, and histopathology of glioblastoma, as well as to compare treatment efficacy in patients treated with selected potential antitumor drugs.

**Abstract:**

Glioblastoma is the most common histologic type of all gliomas and contributes to 57.3% of all cases. Despite the standard management based on surgical resection and radiotherapy, it is related to poor outcome, with a 5-year relative survival rate below 6.9%. In order to improve the overall outcome for patients, the new therapeutic strategies are needed. Herein, we describe the current state of knowledge on novel targeted therapies in glioblastoma. Based on recent studies, we compared treatment efficacy measured by overall survival and progression-free survival in patients treated with selected potential antitumor drugs. The results of the application of the analyzed inhibitors are highly variable despite the encouraging conclusions of previous preclinical studies. This paper focused on drugs that target major glioblastoma kinases. As far, the results of some *BRAF* inhibitors are favorable. Vemurafenib demonstrated a long-term efficacy in clinical trials while the combination of dabrafenib and trametinib improves PFS compared with both vemurafenib and dabrafenib alone. There is no evidence that any MEK inhibitor is effective in monotherapy. According to the current state of knowledge, *BRAF* and MEK inhibition are more advantageous than *BRAF* inhibitor monotherapy. Moreover, mTOR inhibitors (especially paxalisib) may be considered a particularly important group. Everolimus demonstrated a partial response in a significant proportion of patients when combined with bevacizumab, however its actual role in the treatment is unclear. Neither nintedanib nor pemigatinib were efficient in treatment of GBM. Among the anti-VEGF drugs, bevacizumab monotherapy was a well-tolerated option, significantly associated with anti-GBM activity in patients with recurrent GBM. The efficacy of aflibercept and pazopanib in monotherapy has not been demonstrated. Apatinib has been proven to be effective and tolerable by a single clinical trial, but more research is needed. Lenvatinib is under trial. Finally, promising results from a study with regorafenib may be confirmed by the ongoing randomized AGILE trial. The studies conducted so far have provided a relatively wide range of drugs, which are at least well tolerated and demonstrated some efficacy in the randomized clinical trials. The comprehensive understanding of the molecular biology of gliomas promises to further improve the treatment outcomes of patients.

## 1. Introduction

In 2020, GLOBOCAN 2020 reported an estimated 19.3 million new cancer cases diagnosed and roughly 10 million cancer deaths globally. Statistically, cancer of the brain and nervous system is the 21st most common malignancy and is responsible for 308,102 new cases and 251,329 deaths worldwide annually [1]. Approximately 5–6 patients out of every 100,000 people are diagnosed with primary malignant brain tumors annually [2]. Gliomas are the most frequently occurring type of primary intracranial tumor that constitute 31% of all brain and central nervous system (CNS) tumors, and as well as 81% of all malignant brain and CNS tumors [3,4]. Gliomas originate from the supporting cells called neuroglial, in which there can be distinguished astrocytes, ependymal cells, and oligodendroglial cells [5]. The morbidity rate of gliomas oscillates from 4.67 to 5.73 per 100,000 people. Moreover, in the United States, gliomas are most often found among non-Hispanic whites, Asian/Pacific Islanders, and American Indians/Alaska Natives [3,6]. Worldwide statistics emphasize that morbidity rate of glioma varies geographically. The number of incidences in America and Northern Europe is approximately two times higher than in Asia [7]. The frequency of glioma soars up with age, whilst the median age at diagnosis in the United States is 65 years [8]. In the United States, glioma occurs with a male-to-female ratio of 1.3:1. Furthermore, over the last 20 years, the number of new cases has remained constant [7]. Exposure to ionizing radiation is a risk factor that significantly predisposes one to the development of gliomas. Inversely, recent studies show that history of allergies or atopic diseases lead to a reduction of glioma development [3]. Rare genetic syndromes such as neurofibromatosis type 1, tuberous sclerosis, and Li–Fraumeni syndrome are correlated with increased risk of glioma development. Furthermore, family history of cancer elevates the risk of glioma two-fold [3,9].

Glioblastoma (GBM) is the most common histologic type of all gliomas that contributes to 57.3% of all cases. It is also related to poor outcome, with a 5-year relative survival rate below 6.9%. The rest types of glioma include non-glioblastoma astrocytoma (27%), oligodendroglial tumors (9%), ependymoma (2%), and gliomas not otherwise specified (7%) [7,10]. Regarding histopathology, gliomas can be divided into two main groups: low-grade gliomas (LGG) and high-grades gliomas (HGG). The most advantageous treatments for LGG are maximal resection with eventual reoperation and adjuvant radiation and/or chemotherapy. On the other hand, optimal medical treatments in HGG include maximal resection and mandatory postoperative chemoradiation [11,12]. These procedures may extend a patient’s overall survival (OS), even over 10–15 years on average in LGG and 14 months in HGG, respectively [13,14]. This unfavorable outcome in HGG is mainly associated with complex molecular pathways and genetically determined signaling cascades [15]. Additionally, gliomas lead to a significant decrease in quality of patients’ life, as well as those of their relatives [16]. Firs-line medical treatment recommended so far by clinicians is still insufficient and may require modifications in the future [17]. A comprehensive understanding of the molecular biology of gliomas would allow developing more effective targeted therapies [11].

## 2. Current Treatment Guidelines in Advanced Glioblastoma

Basing on a multidisciplinary approach, treatment of the GBM depends on its clinical staging. Primarily (if it is feasible), the first-choice procedure includes surgery. The aim of maximal safe resection is histological evaluation to make a proper diagnosis, clinical genotyping, and volume reduction of a tumor. Unfortunately, total resection of GBM is almost impossible due to its diffusive characteristic into adjacent tissues and its unfavorable location in the eloquent areas of the brain [18,19,20]. In order to reduce the risk of new postoperative neurological deficits, during the surgery, neuronavigation systems, intraoperative image studies with magnetic resonance imaging (MRI) or ultrasounds, fluorescence dye 5-aminolevulinic acid (5-ALA), and cortical/subcortical mapping techniques may be used [21]. Extensive in-filtration into the brain in GBM prompts the consideration of additional therapies. Studies indicate that chemotherapy and radiation therapy result in prolonged survival in metastatic GBM [2,22]. According to the European Association of Neuro-Oncology (EANO), radiotherapy combined with temozolomide chemotherapy (75 mg/m^2^ daily throughout radiotherapy, including at weekends) plus six cycles of maintenance temozolomide (150–200 mg/m^2^, 5 out of 28 days) is the recommended line of treatment for newly diagnosed GBM in adults under 70 years, who are in good general and neurological condition measured in Karnofsky performance status (KPS) score.

Moreover, the standard of care for patients aged under 70 years with low KPS score and for patients aged over 70 years with O 6-methylguanine-DNA methyltransferase (*MGMT*) promoter, non-methylated, remains hypofractionated radiotherapy (such as 40 Gy in 15 fractions). *MGMT* promoter methylation testing is crucial because patients aged over 70 years, with *MGMT* promoter methylated, should be treated with temozolomide alone. Palliative care is the only currently available option for patients with very unfavorable prognostic factors that justify exclusion from other treatment [23]. Evidence indicates that the median OS in patients in good general condition who received radiotherapy combined with temozolomide reaches 14.6 months in comparison with 12.1 months for radiotherapy alone [19]. Antitumor properties of temozolomide were demonstrated in 1987, but its efficacy in the treatment of patients suffering from GBM was approved by the Food and Drug Administration (FDA) in 2005. Since then, temozolomide as an oral alkylating agent is widely administered in GBM. Compared with the other chemotherapeutic such as lomustine, procarbazine, or vincristine, it is characterized by better safety profile, longer progression-free survival (PFS) period, and improved OS [24,25]. However, in the case of glioblastoma, the phenomenon of drug resistance is also a key problem [26]. This also concerns temozolomide. The vast majority of patients develop resistance to its activity. After a PFS period of 7 to 10 months, the recurrence of GBM is almost inevitable [27]. Tumor-treating fields (TTFields) is a novel and effective adjuvant treatment option for newly diagnosed patients, which can be given in combination with maintenance temozolomide chemotherapy. Nevertheless, there is a strong concern about the feasibility and cost-effectiveness of this modality [28,29].

In the randomized, open-label trial, including 695 patients with glioblastoma after surgical resection (or biopsy) and concomitant radiochemotherapy the component of TTFields to maintenance temozolomide chemotherapy and to maintenance temozolomide alone, resulted in statistically significant improvement in progression-free survival and overall survival. Patients were randomized 2:1 to TTFields plus maintenance temozolomide (150–200 mg/m^2^; 6–12 cycles) chemotherapy (*n* = 466) or temozolomide alone (*n* = 229), while the TTFields consisted of low-intensity, 200 kHz frequency, alternating electric fields, delivered via transducer arrays on the head. Progression-free survival was tested at α = 0.046. The secondary end point was overall survival (tested hierarchically at α = 0.048). Median overall survival was 20.9 months in the TTFields-temozolomide group and 16.0 months in the temozolomide-alone group (HR, 0.63; 95% CI, 0.53–0.76; *p* < 0.001). Systemic adverse event frequency was 48% in the TTFields-temozolomide group and 44% in the temozolomide-alone group. In May 2009, the FDA approved bevacizumab as a single agent for recurrent GBM if prior modalities of therapy failed. As a result, bevacizumab has increasingly been used in patients with recurrent GBM, but it is not indicated as first-line treatment in case of newly diagnosed GBM. Nowadays, there are studies conducted under its combinations with concomitant cyto-toxic and immune therapies [30]. In order to improve the overall outcome for patients with GBM, the new therapeutic strategies are needed. Molecular targeted therapy and immuno-therapy, which have been already used in a wide range of malignancies, constitute a promising treatment option for GBM. Advances in recent years let to comprehensive understanding of genomic and proteomics GBM landscape. Taking into consideration the fact that novel therapeutics and conventional treatment affect different targets, synergistic or combined therapy may be advantageous [31].

## 3. Main Pharmacological Options in the Treatment of Glioblastoma

Gliomas are the most frequent primary brain tumors that can be divided into two main sub-groups: diffuse gliomas and gliomas showing a more circumscribed growth pattern (‘nondiffuse gliomas’) [32,33]. Historically taking into account similar histological characteristics, gliomas include astrocytomas, brain stem gliomas, ependymomas, oligodendrogliomas, optic pathway gliomas, and mixed gliomas. However, this classification cannot provide information about malignancy of a neoplasm. The development of molecular biology resulted in better understanding of exact mechanisms laying at the basis of gliomas and providing novel therapeutic methods. Therefore, in 2014, the International Society of Neuropathology included molecular features on diagnostic process of brain tumors [34]. As a consequence, in 2016, the World Health Organization Classification of tumors of the central nervous system (CNS WHO) was modified. It led to the development of the new classification of gliomas based on both microscopy (phenotype) and molecular parameters (genotype). The up-dated CNS WHO classifies tumors in four grades (grade I refers to tumors that are slowly growing, non-malignant, associated with long-term survival, whilst grade IV refers to aggressive tumors that reproduce rapidly and are correlated with the poorest prognosis) [35,36]. GBM is the most common and lethal grade IV glioma subtype with one of the worst 5-year OS rate among all human cancers [37,38]. GBMs appear mainly in the brain (the most frequently in cerebral hemispheres), but they can also form in the other locations such as the brain stem, cerebellum, and spinal cord [39,40]. Primarily, GBM was considered to be derived from glial cells; however, data suggest that both embryonic and adult neural stem cells have the ability to develop into cells of GBM. Essentially, a tumor may arise from a wide range of cells at multiple stages of differentiation from stem cell to neuron or glia [18,41].

The analysis of The Cancer Genome Atlas revealed that among thousands of genes involved in the development of GBM, there are 426 up-regulated genes and 65 down-regulated genes described as significant due to its correlation with patient’s survival [42,43]. According to the researches, GBMs occur as a result of gene–environment interactions [44]. The leading cause of GBM development is critical signaling disruption that is often related with mutations targeted diverse kinases. Consequently, some kinase inhibitors are considered as promising novel therapeutic options. Regarding pathogenesis of GBM, most important kinases include epidermal growth factor (*EGFR*), platelet-derived growth factor (*PDGFR*), hepatocyte growth factor (*MET*), fibroblast growth factor (*FGFR*), vascular endothelial growth factor (*VEGFR*), and insulin-like growth factor 1 receptor (*IGF1R*) [45]. Data show that there are PI3K signaling cascades, and the MEK/ERK pathway as well, involved not only in enhancing the survival of GBM, but also in conferring resistance against radio- and chemotherapy [46]. Moreover, *BRAF* mutation, which appears in approximately 8% of GBMs, activates the RAF–MEK–MAPK signaling cascade. It leads to cell proliferation, inhibition of apoptosis, and tumor growth [47,48]. Altered cellular metabolism plays a crucial role in the cancer’s progression. There are also patterns that determine the progression of GBM. Recent studies describe isocitrate dehydrogenase (*IDH*) enzyme as a marker of good prognosis in GBM. So far, even 95% of GBs are IDH-wild type (associated with poor outcome) [49]. Knowledge about molecular alterations in GBM is used to diagnose and classify tumors. Moreover, it allows us to choose the best treatment for the patient [50]. Despite increasing knowledge of GBM molecular biology, the standard of care remains a radical surgical resection (if feasible) combined with adjuvant radiotherapy and chemotherapy with temozolomide. However, in most cases this treatment is related with relapse, progression, and poor OS [47]. For the purpose of developing new treatment strategies, there is a need to recognize mutations or pathways involved in GBM and particular potential therapeutic targets [51]. Innovative clinical trials, related to immunotherapy and targeted therapies, hold a new hope for improving outcomes in patients with GBM [31].

### 3.1. BRAF in Glioblastoma

The v-raf murine sarcoma viral oncogene homolog B (*BRAF*) gene-activating mutations appear in 15% of young adults suffering from GBM [47,52]. It is located in the long arm of chromosome 7 at position 34. Missense mutations and aberrant fusions are the most common reasons of *BRAF* alterations. Among missense mutations, the most frequent remains the substitution of valine for glutamate at position 600 (V600E), whilst among aberrant fusions dominates aberration in the N-terminal portion of the protein [53]. Moreover, based on the newly proposed classification, *BRAF* mutations can be divided into three classes: class-1 (kinase-activated, codon 600 including V600E, V600M, V600K, V600R, V600D mutations), class-2 (kinase-activated, non-codon 600 associated with K601E, K601N, K601T, and L597Q mutations) and class-3 (kinase-impaired related to D954N, N581S, G466V, D594G, G466E, and G596D mutations). However, only *BRAF* V600 mutants (that belong to class I) are sensitive to currently available *BRAF* inhibitors [48,54]. The serine/threonine-protein kinase, B-Raf (encoded by the *BRAF* gene) is a part of the RAS/RAF/MEK/MAPK signaling cascade that is responsible for downstream signaling pathways of epidermal growth factor receptors (EGFR) or platelet-derived growth factor receptors (PDGFR). The roles of both of them have been confirmed in gliomagenesis and cell proliferation. One of the crucial mutations in *BRAF* is the V600E mutation that is related to increased kinase activity and oncogenic stimulation. Its prevalence is characteristic for pediatric gliomas and oscillates around 50% in children and young adults with epithelioid GBM. GBMs with BRAF mutation differ significantly in location, survival rates, and global gene-expression profiles from the rest of the GBMs. Moreover, there is observed increased expression of genes implicated in MET pathway signaling, protein processing, immune function, and invasion. Due to the conducted study, patients with *BRAF* mutation are usually younger, and they survive longer in comparison with other patients with GBM [50,52].

So far, *BRAF* inhibitors have been approved for the treatment of patients diagnosed with melanoma, non-small cell lung carcinoma, and thyroid cancer [50,55]. In recent years, studies have indicated that *BRAF*-targeted therapy may constitute an effective treatment option for patients with *BRAF*-mutated GBM. Evidence suggests that *BRAF*-targeted therapy is generally well tolerated and leads to rapid clinical improvement [56]. The mechanism of action of *BRAF* inhibitors, such as vemurafenib, dabrafenib, and encorafenib, is based on inactivation of the Raf kinase by competitive occupation of the ATP binding pocket, which leads to the loss of active conformation. Its consequences are disruptions of the downstream MAPK signaling cascade, G1 cell-cycle arrest, and cell apoptosis [57]. As of now, there is no drug in this group approved by the FDA in patients with GBM. Nevertheless, *BRAF* inhibitors such as dabrafenib and vemurafenib are being actively researched and developed (Table 1) [47].

### 3.2. MEK in Glioblastoma

The mitogen-activated extracellular signal-regulated kinase (*MEK*) can be divided into two different forms: *MEK1* and *MEK2*. MEK is a downstream kinase, which is a part of the RAS cascade, which is responsible for signal transduction associated with key cellular processes such as proliferation, survival, and differentiation [80,81]. Therefore, the dysregulation of the Ras–Raf–MEK–ERK pathway is frequently noted in carcinogenesis. It may be activated by both growth factors and alterations of several proteins involved in this cascade. The prevalence of mutations is lowering as one moves further downstream in the pathway, and hence it is the most frequent in *RAS* (22%) and *BRAF* (7%), whilst it is the rarest in *MEK* (less than 1%) and *ERK* (extremely infrequent). Moving on to the beginning of the Ras–Raf–MEK–ERK cascade mechanism, growth factor binds to a tyrosine kinase receptor (TKR) such as epidermal growth factor receptor (EGFR), leading to the receptor’s activation and the subsequent activation of Ras small GTPases. This results in the formation of Ras-GTP, which is followed by Ras activation. Then, Raf serine/threonine kinases constitute downstream effector targets of Ras. Finally, activated Raf proteins in the process of phosphorylation activate MEK1 and MEK2. Moreover, this leads to phosphorylation and the activation of the ERK kinases (ERK1 and ERK2) [82,83]. Hyperactivity of this cascade is significantly correlated with tumor cell proliferation and cancer progression. Therefore, molecular therapies targeting the Ras–Raf–MEK–ERK signaling pathway are actively developed [84].

In approximately 90% of GBM, dysregulated signal traffic through the Ras pathway is observed [85]. MEK1/2 are not only involved in tumor development, but also in the inhibition of apoptosis. Therefore, *MEK1/2* inhibitors have great potential as an attractive targeted treatment option [86]. Evidence suggests that blocking MEK signaling in GB is associated with antiproliferative effects by preventing cells from dividing and a decline in the percentage of cells positive for Ki67. Trametinib, as a *MEK1/2* inhibitor, has been proved to inhibit the Ras–Raf–MEK–ERK pathway and block its downstream extracellular-related kinases. Moreover, it is able to restrict GBM proliferation, migration, and invasion. So far, trametinib monotherapy has been approved by the FDA for melanoma treatment [46,87]. Besides trametinib, other *MEK* inhibitors such as cobimetinib are considered as novel treatment methods for GBM (Table 1). They are generally well tolerated, and their effectiveness can be assessed based on a decrease in tumor size [88]. Currently, in the treatment of V600E *BRAF*-mutated HGG, the combination of trametinib and dabrafenib appears clinically significant (*BRAF* inhibitor) [59].

### 3.3. PI3K in Glioblastoma

The phosphatidylinositol 3-kinase (PI3K) represents a family of lipid kinases activated by a large number of receptor tyrosine kinases. Mammalian Target of Rapamycin (mTOR), a serine/threonine kinase which is able to capture and transduce signals from different stimuli, belongs to the family of PI3K-related protein kinases. The PI3K/mTOR signaling pathway plays an important role in multiple crucial cellular functions, such as cell metabolism, growth, survival angiogenesis, and motility [89]. Furthermore, the abnormal activity of that pathway is also involved in the development of GBM. *PI3K* mutations are observable in one of four patients with GBM [90]. Most common mechanisms related to the overactivation of the PI3K pathway include phosphatidylinositol-4,5-bisphosphate 3-kinase catalytic subunit alpha (*PIK3CA*) mutations, the loss of phosphatase and tensin homolog (*PTEN*) gene function, and the amplification of epidermal growth factor receptor (*EGFR*) gene expression [91]. The mechanism of the PI3K/Akt/mTOR signaling pathway is crucial. PI3K translocates to the plasma membrane and catalyzes phosphatidylinositol 3,4,5-triphosphate (PIP3) production, which activates serine/threonine kinase phosphoinositide-dependent kinase 1 (PDK1) and Akt, respectively. This results in the suppression of apoptosis. Subsequently, activated Akt by mediating protein synthesis activates mTOR (downstream target of PI3K) [92]. Hyperactivation of the PI3K/Akt pathway in GBM induces both the rapid growth of tumors and multidrug resistance. Consequently, the inhibition of PI3K alone or in combination with other targets may provide cell apoptosis and retarded progression of GBM.

PI3K inhibitors may be classified into Pan-PI3K inhibitors (such as BKM120, XL147, PX-866, GDC-0941, GDC-032, BAY 80-6946, ZSTK474, and AMG 511), isoform-selective PI3K inhibitors (including BYL719, MLN1117, CAL-101, GSK2636771, CH5132799, AMG319, and AZD6482) and dual PI3K/mTOR inhibitors (NVP-BEZ235, XL765, GDC-0084, GDC-0980, GSK2126458, PF-05212384, and PQR309) [93]. Moreover, PI3K can be divided into three classes, in which class I PI3K consists of a catalytic subunit (p110) and a regulator subunit [94]. Pan-PI3K inhibitors, as well as isoform-selective PI3K inhibitors, suppress the activities of p110 catalytic isoforms, whilst dual PI3K/mTOR inhibitors work against both p110 and mTOR complex 1/2. In cancer treatment, there have been over 50 PI3K inhibitors discovered, but currently only several are considered in clinical trials (Table 1) [93]. As evidenced so far, PI3K inhibitors with brain-penetration ability include NVP-BEZ235, XL765, GDC-0084, and PQR 309. Development of the targeted therapies focused on the PI3K signaling pathway would allow for its wide use and clinically significant applications in the treatment of GBM [95].

### 3.4. FGFR in Glioblastoma

Fibroblast growth factors (*FGFs*) are crucial regulators of tissue development, metabolism, differentiation, and repair. FGFs trigger signaling by acting through tyrosine kinase receptors, known as fibroblast growth-factor receptors (*FGFRs*). In the family of FGFRs, there can be four transmembrane receptors distinguished: FGFR 1–4 [96]. FGF – FGFR induces cell signaling pathways such as RAC/JNK, RAS–MAPK (both related to cell proliferation), and PI3K/AKT (associated with cell survival). Moreover, the analysis of gene expression revealed the correlation between somatic mutations of *FGFRs* and GBM progression [97]. With an approximate prevalence of about 6%, *FGFR* genomic alterations are not very common in GBMs [98]. *FGFR3–TACC3* fusions have been identified as the most frequent *FGFR* alterations underlying IDH-wild-type GBM [99]. In a wide range of tumor types, it has been proved that FGFR1 is involved in the development of resistance to cytotoxic and hormonal therapies [100,101]. FGFR1 can also modulate the tumor microenvironment and an angiogenic response, resulting in a decrease in GBM radiosensitivity [102]. The *FGFR1* level increases as the tumor progresses, whilst *FGFR2* expression constantly decreases with GBM grade [103]. Diminished expression of *FGFR2* is closely connected with unfavorable prognosis and poorer outcomes in patients [104]. Hance, higher levels of *FGFR2* correlate with reduced proliferation measured by Ki-67 nuclear antigen expression. The fusion of the *FGFR3* and *TACC3* genes that occurs in 3% of GBM leads to the formation of an oncogenic *FGFR3* [97]. In those cases, enhanced oxidative phosphorylation and mitochondrial activity, which play important roles in GBM, are observed [105]. Current studies have revealed that *FGFR4* contributes to viability, adhesion, migration, and clonogenicity of GBM cells. Furthermore, *FGFR4* is described as a predictor of shorter survival in patients with this type of brain tumor [106]. Taking into consideration the abovementioned, the inhibition of FGFR represents a potential therapeutic target for GBM [107]. Selective inhibitors, such as pemigatinib and nintedanib, have already been tested in clinical trials (Table 1).

### 3.5. VEGF in Glioblastoma

The vascular endothelial growth factor (*VEGF*) family includes proteins that are related with specific receptors such as VEGFR-1, VEGFR-2, VEGFR-3, neuropilin-1, and neuropilin-2 [108]. VEGF is a prognostic angiogenic marker that has been confirmed to play a crucial role in GBM biology [109]. Hypoxic and necrotic conditions activate GBM cells and cause the release of pro-angiogenic growth factors such as VEGF [110]. Produced by tumor, stromal, and inflammatory cells of GBM, VEGF stimulates the VEGF receptor, resulting in proliferation, migration, and survival of endothelial cells. It significantly contributes to both the increase of tumor perfusion and the elevation of interstitial pressure. That induces a loss of the blood–brain barrier and the development of vasogenic oedema with mass effect (the leading cause of morbidity for GBM patients) [111]. GBM belongs to the group of the most highly vascularized solid tumors. It is characterized by strong vascular proliferation that leads to the formation of dilated, tortuous, leaky, and highly permeable blood vessels. As a consequence of an abnormal and dysfunctional vasculature system, the limitations in delivery of chemotherapeutics to the tumor mass may be performed. In GBM, there can be two main mechanisms distinguished at the basis of the development of pathological vessels: vasculogenesis and angiogenesis. A typical feature of GBM remains glomeruloid microvascular proliferation [112,113]. The malignancy vasculature is closely linked with GBM development. Therefore, the degree of angiogenesis highly correlates with the prognosis. Currently, antiangiogenic drugs have been investigated as a potentially compelling anti-GBM therapy [110,114]. Bevacizumab is a humanized monoclonal antibody directed against VEGF-A. In 2009, bevacizumab monotherapy, as the only one of VEGF inhibitors, was approved by the FDA in the treatment of recurrent GBM [115]. Anti-angiogenic therapies, which have been assessed in clinical trials, constitute alternative or complementary options to the conventional GBM treatment. Besides bevacizumab, other antiangiogenic drugs such as aflibercept and multiple VGFR inhibitors have been considered in the management of GBM (Table 1) [116].

## 4. Drug-Evaluating Studies

The main potential antitumor drugs are described in the following paragraph. They comprise *BRAF* inhibitors (vemurafenib and dabrafenib), *MEK* inhibitors (trametinib and cobimetinib), PI3K inhibitors (paxalisib), mTOR inhibitors (everolimus), FGFR inhibitors (nintedanib and pemigatinib), VEGF inhibitors (aflibercept and bevacizumab), and VEGFR inhibitors (nintedanib, pazopanib, sorafenib, sunitinib, lenvatinib, apatinib, and regorafenib) (Table 1). Based on recent studies, treatment efficacy of the drugs, measured by overall survival and progression-free survival in patients treated with the selected targeted drugs, was compared.

### 4.1. Vemurafenib

Vemurafenib is a strong and highly selective *BRAF* V600 inhibitor which was approved by the FDA for the treatment of unresectable or metastatic melanoma [117]. Actually, *BRAF* V600 alterations have been identified not only in melanoma, hairy cell leukemia, papillary thyroid cancer, small-cell lung cancer, and colorectal cancer, but also in GBM [118]. The VE-BASKET (NCT01524978), a phase II trial, was a non-randomized, open-label, histology-agnostic basket study for patients with multiple *BRAF* V600E-mutant non-melanoma solid tumors. Patients were divided into seven cohorts (the first six groups enrolled with pre-specified tumors such as non-small cell lung cancer, ovarian, colorectal, and breast cancers, cholangiocarcinoma, and multiple myeloma, while the seventh cohort included other tumors harboring the *BRAF* V600E mutation) [119]. Among them, twenty-four patients suffering from different mutant gliomas of any grade (including eleven patients with malignant diffuse glioma, six with GBM, and five with anaplastic astrocytoma) were treated with vemurafenib 960 mg twice per day continuously in 28-day cycles until the development of the disease progression, withdrawal, or intolerable adverse effects. In total, there were 18 female and 6 male patients enrolled, and their median age oscillated around 32 years. The primary endpoint included unconfirmed objective radiographic response rate at week 8 or first assessment by Response Evaluation Criteria in Solid Tumors (RECIST) version 1.1. Secondary endpoints were objective response rate (ORR), clinical benefit rate (defined as the confirmed complete or partial response of any duration or stable disease lasting a minimum of 6 months), PFS, OS, and safety. Confirmed ORR was different and ranged widely depending on the analyzed group. It reached 25% (95% confidence interval (CI), 10% to 47%) in the overall group and 9.1% (95% CI, 0.2 to 41.3%) in patients with malignant diffuse gliomas. A confirmed clinical benefit rate of 27.3% (95% CI, 6% to 61%) was observed in patients with malignant diffuse glioma, in contrast to 37.5% (95% CI, 18.8% to 59.4%) observed in all assessed patients. The overall median PFS for all patients was 5.5 months (95% CI, 3.7 to 9.6 months) and for malignant diffuse glioma it was 5.3 months (95% CI, 1.8 to 12.9 months). Median OS for all patients was 28.2 months (95% CI, 9.6 to 40.1 months), while it reached about 11.9 months (95% CI, 8.3 to 40.1 months) in patients diagnosed with malignant diffuse glioma. Adverse events occurred in 20% of all enrolled patients. Among them, the most frequent included arthralgia, melanocytic nevus, palmar–plantar erythrodysesthesia, and photosensitivity reaction. Only 13% of patients developed grade 3 and 4 events, and no grade 5 treatment-related events were noted. The VE-BASKET study provided information on the long-lasting antitumor activity of vemurafenib in some patients suffering from *BRAF* V600-mutant gliomas. However, the efficacy of the treatment was highly heterogeneous and was associated with histologic subtype of glioma [58]. The long-term clinical benefits to the targeted therapy of *BRAF* V600-mutant glial and glioneuronal tumors in adult patients were confirmed in a follow-up study in 2021 by Berzero et al. [120].

### 4.2. Dabrafenib and Trametinib

Studies suggest that dabrafenib, as a selective *BRAF* inhibitor, is characterized by a greater brain distribution and penetration compared with vemurafenib [121,122]. According to the available data, dual-targeted therapy with *BRAF* and MEK inhibition is more advantageous than *BRAF* inhibitor monotherapy in the treatment of *BRAF*-mutant cancers. The administration of a *BRAF* inhibitor alone may be associated with the development of resistance and multiple adverse effects (especially skin toxicity). On the other hand, the addition of an MEK inhibitor such as trametinib delays or prevents the development of resistance and mitigates the serious side effects of dabrafenib [123]. Previously, in two phase III trials, it was proven that a combination therapy with dabrafenib and trametinib improves PFS compared with dabrafenib alone (NCT01597908) or with vemurafenib alone (NCT01597908) in patients with *BRAF* mutant metastatic melanoma [124,125]. On 22 June 2022, the FDA granted accelerated approval of dabrafenib in combination with trametinib for the treatment of adult and pediatric patients ≥6 years of age with unresectable or metastatic solid tumors with *BRAF* V600E mutation. That recommendation includes patients who have progressed following prior treatment and have no satisfactory alternative treatment options. The efficacy and safety of dabrafenib in conjunction with trametinib were evaluated in several clinical trials such as BRF117019 (NCT02034110), NCI-MATCH (NCT02465060), and CTMT212X2101 (NCT02124772) [126]. 

Recently, the phase II basket trial Rare Oncology Agnostic Research (ROAR) of dabrafenib plus trametinib in patients with *BRAF* V600E mutation-positive recurrent or refractory HGG (including GBM), and LGG as well, has been on-going. This trial constitutes a part of the study BRF117019 (NCT02034110), which is a multi-cohort, multi-center, nonrandomized, open-label trial in adult patients with selected *BRAF* V600E-mutant tumors. Participants received 150 mg of dabrafenib orally twice daily in combination with 2 mg of trametinib orally once daily until disease progression, unacceptable toxicity, or death. There could be two cohorts (the HGG cohort consisted of 45 patients with 31 GBM patients, and the LGG cohort consisted of 13 patients) distinguished. The primary end-point included investigator-assessed ORR (complete response and partial response in both cohorts and a minor response for LGG). The results showed that in the HGG cohort, an objective response was observed in 33% of patients (95% CI, 20% to 49%), while a complete response was noted in 6.7%. Moreover, among patients with GBM, an objective response was observed in 32% and a complete response occurred in 6.5%. In the HGG cohort, the median duration of response was 36.9 months (95% CI, 7.4 to 44.2 months), median PFS was 3.8 months (95% CI, 1.8 to 9.2 months), and median OS was 17.6 months (95% CI, 9.5 to 45.2 months). Serious adverse events were reported in 33% of patients in the HGG cohort, in which the most frequent were seizure (9%), followed by vomiting, headache (4%), and nausea (4%). Based on the results, it was concluded that dabrafenib in combination with trametinib could potentially be used in the clinical practice for patients suffering from glioma [59]. 

Patients with a *BRAF* V600E mutation were enrolled in the subprotocol H (EAY131-H) of the NCI-MATCH, single-arm, open-label, phase II study (NCT02465060). Adult patients with solid tumors, including gastrointestinal tumors, lung tumors, gynecologic or peritoneal tumors, CNS tumors (including 1 patient with GBM), and ameloblastoma of the mandible, were eligible for the study. Furthermore, participants with melanoma, thyroid cancer, or colorectal cancer were excluded. The diseases were treated with 150 mg of dabrafenib twice per day and 2 mg of trametinib once per day in continuous 28-day cycles until disease progression, intolerable toxicity, or study withdrawal. The primary endpoint of the trial was centrally assessed ORR, while the secondary endpoints included PFS, 6-month PFS, and OS. In total, the ORR of 37.9% (90% CI, 22.9% to 54.9%) and the median duration of response of 25.1 months (90% CI, 12.8 months to NA) were noted in this cohort. The median PFS was 11.4 months (90% CI, 8.4 to 16.3 months), while the 6-month PFS rate was 68.4% (90% CI, 55.4% to 84.4%) and the median OS was 28.6 months. One patient suffering from GBM had a maximal decrease in the sum of measured lesions of 59%. Among the adverse events that occurred in all treated patients, those most common included fatigue (74% of all participants), nausea (57%), fever (51%), chills (54%), and alkaline phosphatase elevation (31%). There was one grade 4 adverse event (sepsis), while grade 5 adverse events were not observable. This trial revealed that *BRAF*/MEK inhibition may be used as an efficient treatment option in the majority of *BRAF* V600-mutated cancers [60]. 

Oral dabrafenib in combination with trametinib is undergoing a clinical trial as a treatment option for pediatric patients with HGG or LGG—a global, open-label, phase II study (NCT02684058). The inclusion criteria of this study include diagnosis of *BRAF* V600 mutant HGG that has relapsed, progressed, or failed to respond to frontline therapy, diagnosis of *BRAF* V600 mutant LGG with progressive disease following surgical excision, or non-surgical candidates with the necessity of beginning their first systemic treatment because of a risk of neurological impairment with progression. In both cases, it was required to confirm measurable disease. Patients were divided into two cohorts (the single-arm *BRAF* V600-mutated HGG cohort treated with dabrafenib twice daily plus trametinib once daily, and the *BRAF* V600-mutated LGG cohort that was randomized 2:1 to receive either dabrafenib in combination with trametinib or carboplatin plus vincristine). In the HGG cohort, the primary endpoint of the study was ORR as determined by the investigator’s assessment based on MRI or CT (computed tomography) scans using Response Assessment in Neuro-Oncology (RANO) criteria. Contrastingly, in the LGG cohort, the primary endpoint was ORR as determined by independent assessment according to the RANO criteria. Duration of response, PFS, time to response, clinical benefit rate, OS, and tolerability of the therapy were taken into consideration as secondary endpoints in both cohorts [127].

### 4.3. Cobimetinib

Cobimetinib is a highly selective MEK inhibitor that was approved in combination with vemurafenib in 2015 for the treatment of patients with unresectable or metastatic *BRAF* V600E/V600K-mutated melanoma [128]. Dysregulation of the MAPK pathway is implicated in oncogenes is of a number of pediatric tumors, including gliomas. Recently, the multicenter phase I/II trial of cobimetinib in pediatric and young adult patients with relapsed or refractory solid tumors (iMATRIX-cobi) was ongoing. Of 56 enrolled patients treated with cobimetinib monotherapy, 18 received tablets and 38 received suspension. The most common solid tumor among eligible patients was LGG (32 cases), while HGG was diagnosed in 5 participants. The primary endpoint of the study was safety, and the secondary endpoints included pharmacokinetics and antitumor activity. Histologically or cytologically confirmed tumors with known/expected MAPK pathway involvement constituted the eligibility criteria. Other eligibility criteria included measurable/evaluable disease according to the International Neuroblastomas Response Criteria (INRC), RANO for HGG, or RECIST; availability of tumor tissue; Lansky or KPS ≥ 50% for children aged <16 or ≥16 years, respectively; life expectancy ≥3 months; and body weight ≥20 kg. The results showed that adverse events of special interest developed in 45% of enrolled patients, in which the most common were epistaxis, increase of aspartate aminotransferase, and elevated creatine phosphokinase. However, of all identified treatment-related adverse events, the most frequently reported were gastrointestinal and skin disorders. No complete responses were noted. Overall, 5% of eligible patients had a partial response and in 59% of participants a stable disease was observable. Moreover, in the vast majority of patients diagnosed with HGG, disease progression occurred. Overall, the response rate was 5.4%; PFS was 14.8 months (95% CI, 3.6 to 14.8), and the median OS was not reached. Due to unfavorable responses, the criteria to continue cohort expansion were not met. Despite being well tolerated and potentially advisable in tumors with MAPK pathway involvement, cobimetinib did not show sufficient efficacy in pediatric and young adult patients [61].

### 4.4. Paxalisib

Paxalisib (GDC-0084) is a small molecule with the ability to penetrate the blood–brain barrier and to inhibit the PI3K/AKT/mTOR pathway. In 2020, The FDA granted a fast-track designation to paxalisib for the treatment of patients with newly diagnosed GBM with unmethylated *MGMT* promotor status who received initial chemoradiotherapy with temozolomide [129]. Previously, in an open-label, multicenter, phase 1, dose-escalating study (NCT01547546), it was proven that paxalisib is reasonably well-tolerated in patients with progressive or recurrent HGG [130]. Currently, paxalisib monotherapy (administered orally in 28-day cycles until disease progression or unacceptable toxicity) as a possible first-line treatment for patients with newly-diagnosed GBM was assessed in an open-label, multicenter, dose-escalation, phase 2 study (NCT03522298). Paxalisib safety, tolerability, recommended phase 2 dose, pharmacokinetics, and clinical activity were evaluated in patients with newly-diagnosed GBM with unmethylated *MGMT* promoter status as adjuvant therapy following surgical resection and chemoradiation with temozolomide. The primary endpoints included safety and tolerability of paxalisib, while the secondary endpoints were pharmacokinetic parameters, OS, and PFS. A total of 30 patients (70% males, 83.3% white, mean age 58.5 years) were enrolled in this study. The vast majority of participants were given between 1 and 6 treatment cycles. Paxalisib, 60 mg, once daily was recognized as the maximum well-tolerated dose. The results showed that the median OS was 15.7 months (11.1 to 19.1) and the median PFS oscillated around 8.4 months (6.6 to 10.2). The safety profile was favorable and the most common side effects included hyperglycemia, oral mucositis, skin rash, and fatigue. The identified adverse events were consistent with other PI3K inhibitors. Based on the results, we can conclude that the maximum tolerated dose was associated with encouraging clinical activity, prolonged PFS, and improved OS. Therefore, paxalisib is a promising drug for treating GBM [62]. 

### 4.5. Everolimus

Everolimus (RAD001), a derivative of rapamycin, is a proliferation signal inhibitor in mTOR. Due to its antitumor activity, the FDA approved everolimus for treatment of multiple malignancies such as renal cell and breast cancer, pancreatic neuroendocrine tumors, and subependymal giant-cell astrocytomas associated with tuberous sclerosis. Nowadays, everolimus is considered as a therapeutic agent in the treatment of GBM. Chinnaiyan et al. described a randomized phase II study, RTOG 0913, of everolimus in combination with chemoradiation in newly diagnosed GBM (NCT01062399). Enrolled patients were randomized to receive either standard therapy (temozolomide and radiation therapy) or standard therapy combined with daily everolimus (10 mg). The primary endpoint of the study was PFS, and the secondary endpoints included OS and treatment-related adverse events. The median evaluation time frame achieved was 27.7 months. Median PFS for the control arm was 10.2 months (95% CI, 7.5 to 13.8 months), compared with 8.2 months (95% CI, 6.5 to 10.6 months) for participants randomized to receive temozolomide and radiation therapy in conjunction with everolimus. Moreover, the median OS in patients treated with the standard therapy was 21.2 months (95% CI, 16.6 to 29.9 months); while, in patients who additionally were given everolimus it was 16.5 months (95% CI, 12.5 to 18.7 months). Treatment-related grade 3–5 toxicities were observed more frequently in patients who received everolimus (80%) than in patients treated with the standard therapy (42.3%). The most common grade 4 everolimus-associated adverse events included bone marrow suppression manifested by lymphopenia and thrombocytopenia. The RTOG 0913 study showed that due to poor outcomes and unfavorable safety profiles, combination treatment with everolimus, radiation therapy, and temozolomide is not related to advantages over the standard therapy in patients with newly diagnosed GBM [63]. 

Furthermore, a single-arm, phase II study, NCCTG N057K, also assessed the combination of the everolimus with conventional temozolomide-based chemoradiotherapy in patients with newly diagnosed GBM. Oral everolimus (at the dose of 70 mg once per week) was started 1 week prior to radiation and temozolomide, followed by adjuvant temozolomide. Patients received everolimus until disease progression. The primary endpoint was OS at 12 months, and secondary endpoints included toxicity and time to progression. In total, the responses in 104 eligible patients were evaluated. The evaluation time frame was 17.5 months, patients had OS at 12 months of 64%, a median OS of 15.8 months (95% CI, 13.0 to 20.3), and a median PFS of 6.4 months (95% CI, 5.4 to 9.0). Grade 3 or 4 hematologic adverse events (with the predominance of thrombocytopenia and neutropenia) developed in 25 patients, while grade 3 or 4 nonhematologic events (such as fatigue or hypercholesterolemia) occurred in 45 patients. Therapy with everolimus had moderate toxicity compared with historical trials and was not related to prolonged OS in patients with newly diagnosed GBM [64]. 

The aim of the phase II study (NCT00805961) was to evaluate the efficacy of adding bevacizumab and everolimus to the standard radiation therapy combined with temozolomide in the first-line treatment of patients diagnosed with GBM. Following surgical resection or biopsy, patients with newly diagnosed GBM received standard radiation therapy combined with temozolomide plus bevacizumab 10 mg/kg intravenously every 2 weeks. Subsequently, 4 weeks after the last dose of radiation, participants were treated with orally administered everolimus (10 mg daily) and continued bevacizumab every 2 weeks. Enrolled patients received therapy until GBM progression or unacceptable toxicity occurred. The primary endpoint of the study was PFS, and the secondary endpoints included ORR and OS. Of the 68 eligible patients, 56 had previously performed surgical resection or debulking of the GBM. After combination therapy, 29 participants (57%) showed partial responses. The median PFS was 11.3 months (95% CI, 9.3 to 13.1 months) and the median OS was 13.9 months (95% CI, 12.4 to NA months). Maintenance treatment with bevacizumab and everolimus was generally well-tolerated. Fatigue (27%), pneumonitis (7%), stomatitis/mucositis (5%), and hyperlipidemia (4%) were the most frequent grade 3/4 adverse events associated with everolimus administration. The addition of bevacizumab and everolimus to the standard therapy appeared to be feasible and efficacious. However, there is a need to provide phase III studies to clarify the role of bevacizumab and everolimus in the treatment of GBM [65]. 

### 4.6. Nintedanib

Nintedanib (BIBF1120) is an oral, small-molecule intracellular inhibitor that effectively targets several tyrosine kinases such as FGFR1-3 and VEGFR1-3. It presents anti-angiogenic and anticancer properties. Moreover, nintedanib is believed to have great potential to overcome the problem of resistance to prior anti-VEGF therapy [131]. For the purpose of assessment of this hypothesis, a phase II, single-arm, open label clinical trial (NCT01380782) was conducted in adult patients with first or second recurrence of GBM, stratified by prior treatment with bevacizumab. Thirty-six patients were enrolled and divided into two arms: Arm A (bevacizumab-naive) and Arm B (bevacizumab-treated), which consisted of twenty-two and fourteen patients, respectively. All participants received 200 mg nintedanib twice a day orally for a 28-day cycle until disease progression or intolerable adverse events occurred. The primary objective of the study was to determine 6-month PFS in bevacizumab-naive participants with re-current GBM and to validate 3-month PFS in bevacizumab-treated patients with recurrent GBM. Secondary objectives in both arms were determining OS, radiographic response rate, time-to-progression, and safety of treatment. Arm A and Arm B included 12 and 10 patients with GBM, respectively. Moreover, in Arm A, four patients (33%) with GBM had stable disease, while in Arm B, one patient (10%) achieved stable disease. In Arm A (GBM only), OS was 6.9 months (95% CI, 3.7 to 8.1 months) and PFS was 0.9 months (95% CI, 0.9 to 2.8 months). In Arm B (GBM only), OS was 2.6 months (95% CI, 1.0 to 6.9 moths) and 0.9 months (95% CI, 0.7 to 0.9 months). The therapy was generally well-tolerated. Mild diarrhea, nausea, vomiting, abdominal pain, and elevated alanine aminotransferase levels were identified as the most common adverse events associated with nintedanib administration. Poor outcomes noted in this trial suggested that nintedanib is not efficient in the treatment of recurrent GBM [66].

### 4.7. Pemigatinib

Pemigatinib is a potent and selective FGFR1-3 inhibitor approved by the FDA for use in locally advanced or metastatic cholangiocarcinoma with an *FGFR2* fusion or rearrangement. In various malignancies, it is explored as a prospective treatment modality. The case report of a patient diagnosed with pilocytic astrocytoma, who was enrolled in an ongoing dose-escalation phase I/II clinical trial of pemigatinib in advanced malignancies with *FGFR* mutations, suggested that intracranial antitumor activity of pemigatenib should be explored in higher-grade gliomas including GBM [132]. Recently, an open-label, phase II, monotherapy study of pemigatinib, FIGHT-209 (NCT05267106), in patients with previously treated recurrent GBM or other primary CNS tumors with an activating *FGFR1-3* mutation or fusion/rearrangement, is ongoing. The aim of the study is to evaluate overall response rate based on RANO or RECIST version 1.1 (primary endpoint) and to determine disease control rate, duration of response, PFS, OS, and treatment safety related to observable adverse events (secondary endpoints). The study is currently recruiting [67].

### 4.8. Aflibercept

Aflibercept is a recombinant fusion protein that can bind vascular permeability factors such as VEGF-A and VEGF-B. In 2019, the FDA approved aflibercept injection for the treatment of diabetic retinopathy [133]. Aflibercept, as a promising therapeutic agent with antivascular properties, was used in trials in patients with recurrent malignant glioma. Patients with histologically confirmed GBM, gliosarcoma, or anaplastic glioma who had received chemoradiotherapy and no more than one adjuvant temozolomide-containing regimen and whose disease had progressed were enrolled in a phase II study evaluating the efficacy of aflibercept (NCT00369590). The primary end point for patients with GBM was a 6-month (26-week) PFS rate. Moreover, in this trial, secondary end points included radiographic response rate, time to progression, OS, and toxicity. Modified MacDonald criteria were used as a tool to assess the response to treatment. Administered intravenously, the dose of aflibercept was 4 mg/kg on day 1 of every 14-day cycle. There were a total of 58 patients, consisting of 16 patients with anaplastic glioma and 42 patients with GBM or gliosarcoma (including 39 patients diagnosed with GBM). The results showed that three patients (7.7%) with GBM had PFS of more than 6 months. Therefore, the primary endpoint for the GBM cohort was not met. In patients suffering from GBM, the median PFS was 3.0 months (95% CI, 2.0 to 4.0 months) and the median OS was 9.8 months. A partial response was seen in seven (18%) patients suffering from GBM. Evaluating safety, the treatment was moderately well-tolerated. Both grade 3 (35 events) and grade 4 (3 events) toxicities occurred among the participants. On the other hand, grade 5 toxicities were not noted. Undesirable effects (including mainly fatigue, thrombo-embolic complications, wound-healing complications, and CNS ischemia) that necessitated discontinuing aflibercept treatment developed in six patients with GBM. That study, for the first time, considered aflibercept as a treatment approach for patients with recurrent GBM. As the primary endpoint of the trial was not met, it was concluded that aflibercept monotherapy presents minimal activity in recurrent GBM [68]. 

Preclinical models significantly supported a potentially favorable profile of aflibercept in combination with radiotherapy compared with radiotherapy alone or aflibercept alone [134]. A three-arm, phase I study (NCT00650923) enrolled 59 patients with newly diagnosed HGG (including 51 with GBM and 8 with anaplastic glioma) to evaluate the combination of aflibercept with radiotherapy and temozolomide. The primary endpoint was safety and determination of the maximum tolerated dose of aflibercept in each of the three arms. In the trial, the maximum tolerated dose of aflibercept was defined as 4 mg/kg every 2 weeks and the therapy was moderately well tolerated. Incidences of treatment interruption occurred in all participants. The main reasons for therapy discontinuation were disease progression (47%) and toxicities (36%). The most common treatment-related adverse events with aflibercept and temozolomide included lymphopenia, neutropenia, thrombocytopenia, seizures, fatigue, and hypertension. The study met its primary endpoint; however, the subsequent study was not continued due to the moderate toxicity and unfavorable efficacy [135].

### 4.9. Bevacizumab

Bevacizumab, as a VEGF-A inhibitor, is the first drug of the anti-VEGF drug group approved by the FDA as a treatment option for recurrent GBM in adults. The recommended dosage in that indication is 10 mg/kg intravenously every 2 weeks [136]. However, a specific dose of bevacizumab calculated based on VEGFA expression levels showed an increased antitumor effect in cell cultures [137]. Currently, many clinical trials are being conducted involving bevacizumab. One of them, described by Wick et al., is a phase III, multicenter, randomized (2:1), open-label study, EORTC 26101 (NCT01290939), exploring the efficacy and safety of this agent combined with lomustine in patients with recurrent GBM. Histologic confirmation of GBM with unequivocal first progression after standard chemoradiotherapy with temozolomide at least 3 months after the end of radiotherapy constituted including the criteria of the trial. Other eligibility criteria, such as WHO performance status ≤ 2 and adequate hematologic, renal, and hepatic function, were also required. In the case of surgery, it should have been performed at least 2 weeks before randomization, whilst the study treatment could have begun over 28 days after the last surgical procedure. Patients were randomized (2:1) to be given bevacizumab (10 mg/kg every 2 weeks) with lomustine (90 mg/m^2^ every 6 weeks) or lomustine (110 mg/m^2^ every 6 weeks) monotherapy until GBM progression or unmanageable toxic effects occurred. WHO performance status (0 vs. >0), steroid use (yes vs. no), the largest malignancy diameter (≤40 vs. >40 mm), and institution were used as stratification factors in randomization. The primary endpoint of the study was the OS, defined as the time from randomization to death. Secondary endpoints included investigator-assessed PFS and ORR per the modified RANO criteria, health-related quality of life (HRQoL), cognitive function, and corticosteroid use. In the lomustine monotherapy group (34% of all involved patients), a median of one cycle of lomustine (range, 1 to 8) was administered. Those who represented the combination treatment group (66% of all involved patients) received a median of three cycles of lomustine (range, 1 to 8) and three cycles of bevacizumab (range, 1 to 16). In both groups, the leading cause of study interruption was disease progression. The median age of patients was 57 years and 24.8% of patients were over 65 years old. Male predominance was noted (61%). According to the statistics, 66% of the included patients presented a WHO performance status score > 0, whilst in 56% of cases the largest tumor diameter did not exceed 40 mm. The results of the clinical trial showed that the use of bevacizumab combined with lomustine in patients with recurrent GBM was associated with an objective response in 41.5% of patients, in contrast to OR at the level of 13.9% in the lomustine monotherapy group. The median PFS was 4.2 months (95% CI, 3.7 to 4.3 months) in the combination group, whilst the median PFS reached about 1.5 months (95% CI, 1.5 to 2.5 months) in the monotherapy group. Moreover, the study revealed that the median OS was 9.1 months (95% CI, 8.1 to 10.1 months) in patients who received lomustine with bevacizumab, in comparison to 8.6 months (95% CI, 7.6 to 10.4 months) in patients treated with lomustine as a monotherapy. In total, 9.5% of patients who received the lomustine monotherapy and 38.5% of patients who were given bevacizumab with lomustine developed serious treatment-related adverse events. Pulmonary embolism, arterial hypertension, and hematologic toxic effects were the most common observable adverse effects. Based on the results, no differences in OS were observed between the two treatment groups. Moreover, combination therapy was related to increased toxicity and deterioration in social functioning. The addition of bevacizumab to lomustine was associated with longer median PFS in comparison with the lomustine monotherapy. However, EORTC could not confirm the conclusion of phase II trials, i.e., that the combination treatment based on the administration of lomustine with bevacizumab improves survival in patients suffering from progressive GBM [69]. 

Friedman et al. described a phase II, multicenter, open-label, noncomparative study, AVF3708g (NCT00345163), focused on the efficacy and safety of intravenous administration of 10 mg/kg bevacizumab monotherapy or in combination with 340 mg/m^2^ or 125 mg/m^2^ irinotecan (with or without concomitant enzyme-inducing antiepileptic drugs, respectively) once every two weeks. A treatment cycle lasted 6 weeks and patients in both groups received therapy for 104 weeks or until progression of the disease or planned treatment disruption occurred. Primary endpoints included 6-month PFS and ORR, as determined by independent radiology review. Secondary endpoints were safety and OS. Confirmed GBM in first or second relapse and disease progression confirmed by MRI no later than 14 days before the first study treatment were required in eligible patients. Analysis of the results demonstrated that the estimated 6-month PFS rates were 42.6% in patients who received bevacizumab monotherapy and 50.3% in patients who represented the combination treatment group. In contrast, the noted 6-month PFS for salvage chemotherapy and irinotecan alone was approximately 15%. In the conducted study, the median PFS times oscillated around 4.2 months (95% CI, 2.9 to 5.8 months) for the monotherapy group and 5.6 months (95% CI, 4.4 to 6.2 months) for the combination group. Furthermore, the median OS times were 9.2 months (95% CI, 8.2 to 10.7 months) for patients treated by bevacizumab alone and 8.7 months (95% CI, 7.8 to 10.9 months) for those who were given bevacizumab with irinotecan. Serious adverse events occurred in 26.2% of patients in the bevacizumab group and in 43% of patients in the combination group. In both the groups, the most frequently observed adverse events were hypertension, hemorrhage (overall and intracranial as well), wound-healing complications, and venous or arterial thromboembolism. The results revealed significant antitumor activity of bevacizumab administered alone and bevacizumab used in combination with irinotecan in pre-treated patients with GBM in their first or second relapse [70]. 

The purpose of a 2009, phase II study, NCI 06-C-0064E, was to evaluate the single-agent activity of bevacizumab in patients with recurrent GBM. Patients with recurrent GBM after standard treatment, based on fractionated radiotherapy and chemotherapy with temozolomide, were included into the trial. They received bevacizumab at the dose of 10 mg/kg every 14 days on a 28-day cycle. In case of progression, enrolled patients were asked to participate in a companion trial of bevacizumab combined with irinotecan. The primary endpoint of the study was PFS at 6 months, whilst secondary endpoints included MRI and positron emission tomography (PET) response rates and exploratory analyses of its correlation with PFS. In that study, treatment-related serious adverse events (grade 4) were uncommon (8.3%) with the predominance of thromboembolic events. Other frequent toxicities included hypertension, hypophosphatemia, and thrombocytopenia. The results showed that the response rate was 19.6%, the median PFS was 4.0 months (95% CI, 3.0 to 6.5 months), and the median OS was 7.8 months (95% CI, 5.3 to 13.5 months). It was concluded that bevacizumab monotherapy is both well tolerated and associated with anti-GBM activity in patients with recurrent GBM [71]. Based on two phase II, open-label, non-comparative studies (described above: NCI 06-C-0064E and AVF3708g), bevacizumab was approved by the FDA in 2009 for the treatment of recurrent GBM [138]. 

### 4.10. Pazopanib

Pazopanib is a second-generation, multi-targeted tyrosine kinase inhibitor, which presents antiangiogenic features and exhibits good potency against several receptors including VEGFR1, VEGFR2, and VEGFR3. Evaluation of the efficacy and safety of pazopanib in adult patients with recurrent GBM was the aim of a phase II single-arm study, the North American Brain Tumor Consortium Study 06-02 (NCT00459381). In this trial, the eligibility criteria included a minimum age of 18 years old, no more than two prior relapses in medical history, histological confirmation of GBM or gliosarcoma, indisputable tumor progression, KPS ≥ 60, adequate function of bone marrow, liver, and kidneys, normal coagulation tests, and life expectancy >8 weeks as well. Precisely determined intervals between administrations of pazopanib and other agents were also required. Due to the treatment plan, patients received pazopanib 800 mg orally daily on 28-day cycles without interruptions, until disease progression or unacceptable toxicity occurred. The primary endpoints of the study were PFS at 6 months and determination of the safety profile of this drug. Secondary endpoints included the assessment of the efficacy of this drug, expressed by radiographic response, time to progression, and OS. Thirty-five participants with GBM were enrolled in the study. The most common adverse events that occurred during the clinical trial were hypertension (37%), fatigue (34%), and elevated ALT (40%). Four patients (11%) were not able to continue the therapy as a result of toxicities (predominantly due to thrombotic/embolic events). The results revealed a median PFS of 12 weeks (95% CI, 8 to 14 weeks) and median OS of 35 weeks (95% CI, 24 to 47 weeks) for patients with GBM. The best radiographic response was progressive disease in 11 patients (32%) and stable disease in 21 patients (59%). This phase II study showed that pazopanib monotherapy did not prolong PFS in patients suffering from recurrent GBM. However, pazopanib was generally well-tolerated with treatment-related adverse events, similar to other anti-VEGF/VEGFR drugs [72]. Another study, HYPAZ (NCT01392352), considered pazopanib’s involvement in the induction of hypertension in normotensive patients with various advanced solid tumors (including GBM). Participants received pazopanib at the standard dose of 800 mg daily for 12 weeks. The study revealed that administration of pazopanib led to a significant increase in both systolic blood pressure and diastolic blood pressure, respectively, by 12 (95% CI, 4 to 19) mmHg and 10 (95% CI, 5 to 15) mmHg. This trial highlighted the fact that hypertension remains a serious adverse effect associated with VEGF-inhibitors [139]. 

### 4.11. Sorafenib

Sorafenib is an orally active multitargeted tyrosine kinase inhibitor of the VEGFR, PDGFR, and the Ras/Raf signaling pathway. Hainsworth et al. described a multicenter phase II study (NCT00544817) to evaluate the efficacy of sorafenib in the first-line treatment of patients diagnosed with GBM. Following initial surgery resection or biopsy, patients were treated with concurrent radiotherapy (2.0 grays (Gy)/day; total dose, 60 Gy) and temozolomide (at a dose of 75 mg/m^2^ orally daily). Subsequently, a 6-month maintenance therapy with temozolomide (at a dose of 150 mg/m^2^ orally on days 1–5 every 28 days) and sorafenib (at a dose of 400 mg orally twice daily) was conducted. PFS was the primary endpoint, and ORR and OS constituted secondary endpoints. A total of 47 participants were enrolled in the study. Median duration of administration of temozolomide and sorafenib reached 4 months. Moreover, the median OS was 12 months (95%CI, 7.2 to 16 months), while the median PFS was 6 months (95% CI, 3.7 to 7 months). The combination of temozolomide and sorafenib was generally well-tolerated. Most common grade 3/4 sorafenib-adverse events included hand–foot skin reaction (7%), fatigue (7%), and skin rash (4%). Despite that fact, the addition of sorafenib was not related with improved efficacy of treatment in comparison to the standard therapy results in patients with newly diagnosed GBM [73].

The aim of an open-label, non-randomized, single-center phase II trial (NCT00597493) was to evaluate potential therapeutic benefits associated with sorafenib and protracted daily temozolomide in adult patients with GBM. Participants at any recurrence after standard temozolomide chemoradiotherapy received sorafenib (400 mg twice a day) and continuous daily temozolomide (50 mg/m^2^/day). The primary endpoint of the study was PFS at 6 months. Secondary endpoints included safety, toxicity, radiographic response rate, OS and the pharmacokinetics of sorafenib combined with temozolomide in patients who were on and not on concurrently administered CYP3A-inducing anti-epileptics. To be eligible for this trial, patients with histologically confirmed GBM were required to have recurrence following prior therapy, as defined by the Macdonald criteria. Moreover, inclusion criteria were age ≥18 years, KPS ≥ 60%, stable corticosteroid dosing for at least 1 week prior to therapy initiation, and adequate hematologic, renal, and hepatic function, PT/PTT within normal limits, an interval of at least 2 weeks between prior surgical resection, an interval of 4 weeks from prior chemotherapy or investigational therapy, and an interval of 12 weeks from prior radiotherapy. On the other hand, patients were excluded based on prior treatment with sorafenib. Among the 32 participants who were enrolled in this trial, confirmed partial response was observed in only 1 patient (3%), and 16 patients (50%) had progressive disease. The results showed that the median PFS and OS were 1.6 months (95% CI, 1.0 to 2.9 months) and 10.4 months (95% CI, 6.0 to 13.8 months), respectively. Elevation of amylase and lipase, fatigue, infection, erythrodysesthesia, and rash were the most-frequent toxicities. However, overall, the combination therapy was well-tolerated and there were no observed treatment-related deaths. In conclusion, sorafenib administered with temozolomide had a favorable safety profile, but the efficiency of the treatment in recurrent GBM was limited [74]. 

In a phase II NABTT 0502 study, patients with progressive or recurrent GBM received erlotinib (150 mg once daily) and sorafenib (400 mg twice daily) orally until disease progression or unacceptable adverse events occurred. The primary objective of the study was to estimate the OS associated with this combined treatment, while the secondary objective was to determine the radio-graphic response rate, PFS at 6 months, and to assess the toxicities and pharmacokinetics of this therapy. The median OS was 5.7 months (95% CI, 4.5 to 7.9 months) and the median PFS was 2.5 months (95% CI, 1.8 to 3.7 months). In total, 5%, 41%, and 45% of patients had partial response, stable disease, and progressive disease, respectively. The combination therapy was tolerated and the most common grade 3 or 4 adverse events included fatigue (9%), elevation of lipase (7%), diarrhea (2%), and nausea (2%). The hypothesis that administration of erlotinib and sorafenib would improve the survival time of patients with recurrent GBM was not confirmed [75].

### 4.12. Sunitinib

Sunitinib, an oral tyrosine kinase inhibitor, affects multiple kinase receptors (including VEGFR) that are involved in the progression of several tumors. A phase I clinical dose-escalation study (NCT02058901) conducted in patients with refractory solid tumors (including GBM) showed the promising preliminary antitumor activity of sunitinib. The primary end-points of the trial were to determine the maximum tolerated dose and to evaluate the safety of sunitinib. The maximum tolerated dose was established at 300 mg once weekly and 700 mg once every 2 weeks. It was concluded that sunitinib administration was related to prolonged disease stabilization, tumor-marker response, and the improvement of disease-related symptoms. Moreover, the safety profile was favorable [140]. Recently, van Linde et al. described a pilot study on the determination of tumor concentrations of sunitinib in patients with newly diagnosed GBM (NCT02239952). Restricted delivery of sunitinib through the blood–brain barrier to the GBM tissues has been suggested as a crucial reason for its limited activity. Nevertheless, increased dosage of sunitinib could be related with higher treatment efficacy in patients with GBM [141]. High-dose, intermittent sunitinib is undergoing a multicenter, phase II/III, randomized clinical trial in patients with histologically confirmed de novo or secondary GBM with unequivocal first progression—STELLAR (NCT03025893). Oral sunitinib at a dose of 300 mg administered in a weekly schedule (experimental arm) was compared with lomustine at a dose of 110 mg/m^2^, taken orally on day 1 every 6 weeks (control arm). The primary endpoint of the study was six-month PFS, while the secondary endpoints included OS, objective radiological response rate evaluated according to the RANO criteria, safety, and quality-of-life assessments [76].

### 4.13. Lenvatinib

Lenvatinib is an oral selective inhibitor of VEGFR1-R3 under ongoing trial in patients with previously treated, histologically/cytologically confirmed advanced tumors. An open-label, phase 2, multicohort study, LEAP-005 (NCT03797326), enrolled 187 patients (including 31 suffering from GBM) to receive pembrolizumab 200 mg via intravenous infusion on day 1 of each 21-day cycle (until 35 cycles) in conjunction with 20 mg lenvatinib via oral capsule once a day (until progression of disease or unacceptable toxicity occurred). The primary endpoints of the study were ORR evaluated by RECIST version 1.1 or RANO criteria (for GBM) and safety as well. The secondary endpoints included disease control rate, duration of response, PFS, and OS. In patients with GBM, adverse events graded 3 or more were diagnosed in 35% of patients. In the GBM cohort, 6% of the patients were not able to continue the treatment due to toxicity. ORR in patients with GBM oscillated around 16%, disease control rate was 58%, and median duration of response was 3.2 months. Neither PFS nor OS have been published. The first results of the study were promising; however, the trial remains active, and all cohorts are considered to be expanded [77]. 

### 4.14. Apatinib

Apatinib is a novel, small-molecular, anti-angiogenic inhibitor that selectively targets VEGFR-2 and inhibits tumor growth. Both in vitro and in vivo studies showed that apatinib inhibited the proliferation and malignancy of glioma cells, which is strong evidence of the potential benefit for patients suffering from GBM [142]. Wang et al. described a single-arm, open-label, phase II study (NCT03660761) focused on the efficacy and tolerability of apatinib in combination with dose-dense temozolomide as a first-line treatment in patients with recurrent GBM. Twenty patients with poor responses to radiochemotherapy were enrolled in this trial. Eligibility criteria included a patient age of 18–70 years, KPS of ≥60, histologically confirmed GBM, measurable or evaluable disease by MRI confirmation, and a minimum life expectancy of 8 weeks. Participants were required to be diagnosed with progressive disease (relapse) on MRI defined by RANO criteria after the standard Stupp protocol. Additionally, adequate bone marrow and liver function were compulsory. The primary endpoint of the study was PFS (defined as the time from the start of the study treatment until tumor progression or death), and secondary endpoints included OS, ORR, and DCR. Patients received a combination therapy of apatinib (500 mg orally once daily) and temozolomide (100 mg/m^2^, 7 days on with 7 days off, on a 28-day cycle). Treatment was continued until either disease progression or occurrence of unacceptable toxicity occurred. RANO criteria for HGG, National Cancer Institute, Common Terminology Criteria Adverse Event’s version 4.0 (NCI-CTCAE 4.0), and both Kaplan–Meier curve and log rank test were used to evaluate efficacy, safety, and survival, respectively. In total, 20 patients were enrolled in the study. Based on RANO criteria, one patient had a complete response, eight patients had partial responses, nine patents had stable disease, and in two patients a progression was observed. The results showed that the ORR was 45%, the DCR was 90%, the median PFS for all patients were 6 months (95% CI, 5.3 to 7.8 months), and the median OS was 9 months (95% CI, 8.2 to 12.2 months). The most frequent treatment-related adverse events included hypertension (21%), hand–foot syndrome (16%), leukopenia (14%), and thrombocytopenia (12%). Dose reduction or interruption of apatinib administration led to an immediate decrease in toxicity. Despite the effectiveness and safety of combination therapy with apatinib and temozolomide suggested in this trial, further randomized controlled clinical studies are needed [78]. 

### 4.15. Regorafenib

Regorafenib is an inhibitor of multiple kinases involved in tumor angiogenesis, including VEGFR1–3. So far, the FDA has approved regorafenib for the treatment of metastatic colorectal cancer, gastrointestinal stromal tumors, and hepatocellular carcinoma. Lombardi et al. described a randomized, multicenter, open-label phase 2 trial, REGOMA (NCT02926222), which evaluated the efficacy and safety of regorafenib compared with lomustine in the treatment of recurrent GBM. The aim of the study was to assess the impact of regorafenib on the OS of patients with GBM who progressed after surgery and Stupp regimen with or without bevacizumab. The primary endpoint of the trial was OS, while secondary endpoints included PFS, the proportion of patients achieving disease control and objective response, health-related quality of life, and safety. In total, 119 patients were enrolled in the trial and then randomized to receive regorafenib (59 patients treated by regorafenib 160 mg given as four 40 mg tablets orally once daily for the first 3 weeks of each 4-week cycle) or lomustine (60 patients treated by lomustine 110 mg/m^2^ in 40 mg capsules, up to a maximum dose of 200 mg orally, on day 1 of every 6-week cycle until disease progression, death, or unacceptable toxicity occurred). Median follow up was 15.4 months and 83% of participants died during the clinical trial. The results revealed that median OS was 7.4 months (95% CI, 5.8 to 12.0 months) in the regorafenib group in comparison with 5.6 months (95% CI, 4.7 to 7.3 months) in the lomustine group. The vast majority of patients in both the regorafenib and lomustine groups experienced disease progression or died (95% and 98%, respectively). In each group, 2% of patients had a complete response according to RANO criteria. The median PFS was 2.0 months (95% CI, 1.9 to 3.6 months) in patients who were administered regorafenib and 1.9 months (95% CI, 1.8 to 2.1 months) in those who received lomustine. Of all enrolled patients, 47.9% and 56% of patients treated with regorafenib developed at least one treatment-related grade 3–4 adverse event. In patients who received regorafenib, the most frequent toxicities were hand–foot skin reaction, increased lipase, and increased blood bilirubin. REGOMA revealed that regorafenib therapy was associated with a promising OS benefit in recurrent GBM. Therefore, a subsequent randomized phase 3 study on a larger patient group is considered [79]. Additionally worth mentioning is an ongoing randomized AGILE Trial (NCT03970447) analyzing the efficacy of a variety of therapies in treating patients with newly diagnosed as well as recurrent glioblastoma. Regorafenib is also one of the drugs under observation. The results of this trial may provide valuable insights into its applicability to standard therapy [143].

## 5. Development Prospects of Radiotherapy in Glioblastomas

In accordance with the current version of the NCCN 1.2022 recommendations, the current standard for selecting a radiotherapy regimen in glioblastoma multiform takes into account the patient’s age criterion and takes into account their general condition using the KPS scale. Patients up to 70 years of age and with KPS ≥ 60 are subjected to standard conventional radiotherapy up to a dose of 60 GY fractionated, 2 GY each. In patients up to 70 years of age and with hypofractionated radiotherapy, it can also be combined with simultaneous and possibly adjuvant temozolamide therapy in this group of patients [144]. In patients with KPS ≥ 60 in the presence of large tumor volumes (gliomatosis) and infiltration of the brain stem or spinal cord, the acceptable radiotherapy regimens are 57 GY with a fractionated dose of 1.9 GY and 54-55.8 GY fractionated at 1.8 GY. However, in patients with KPS < 60, hypofractionated radiotherapy involves the use of 40 GY regimens in 15 fractions or 34 GY in 10 fractions, and in elderly patients or those in a poor general condition, when we want a short period of therapy, it is possible to use a dose of 25 GY in 5 fractions [145,146,147].

### 5.1. Re-Irradiation Using Stereotaxic Techniques

Stereotactic radiotherapy in the diagnosis of recurrence in previously irradiated patients is still a challenge for radiotherapists dealing with neoplasms of the central nervous system. In the case of re-radiation of tumor recurrence, the time elapsed since the primary irradiation (optimally >6 months), the volume of the recurrence, the location in relation to the primary irradiation area, and the radiation dose in the critical organs that the patient has already received should be taken into account. Usually, due to the widespread availability, it is preferable to use hypofractionated radiation therapy with IMRT (intensity modulated radiation therapy) up to a dose of 35 GY in 10 fractions or the use of SRS (stereotactic radiosurgery) with a dose of 16 GY in 1 fraction or fractionated stereotactic SFRT (stereotactic fractionated radiotherapy), which most often concerns larger volts at risk [148,149].

The team of Lovo et al., in a published multicenter retrospective analysis, showed that better results in terms of OS were achieved by patients who received SRS 10 months after diagnosis, median 36.2 months vs. 15 months (*p* = 0.004; X 2 = 8.145). With regard to the dose used, it was shown that patients receiving doses higher than 15 GY had a better overall survival, 9 vs. 7 months (*p* = 0.01; X 2 = 6756). Additionally, it was noticed that patients who received self-therapy with bevacizumab and/or chemotherapy after SRS also showed longer overall survival, respectively, 12 vs. 7 months (*p* = 0.04; X 2 = 4196) [150]. In turn, in the analysis by Yaprak et al., it was shown that the use of stereotactic fractionated radiotherapy in the case of recurrence with a median dose of 20 GY (18–30) administered in 2–5 fractions was associated with an extension of OS by 16 months compared with patients who did not use it; the median was 30 vs. 14 months (*p* = 0.001). One-year and two-year OS after SFRT accounted for 48% and 9%, respectively. One-year PFS (progression-free survival) after SFRT was 50% vs. 22% in patients not undergoing reirradiation. Interestingly, this study showed that patients who underwent reoperation before SFRT had worse survival outcomes than patients who underwent SFRT alone (*p* = 0.02) [151]. In a systematic review and meta-analysis of 2095 patients re-radiated using stereotactic techniques, 70% of a 6-month OS and 34% 1-year OS were noted. The 6-month and 1-year PFS were 40% and 16%, respectively [152]. 

An important factor in qualifying patients for re-radiation is the increased risk of radionecrosis. Minitti et al. noticed that the risk for total doses of EQD2 (equivalent dose in 2 GY) < 101 GY did not exceed 3%. With the cumulative dose of EQD2 102–130 GY after SFRT it WAS 7–13% and after SRS and total doses of EQD2 124–150 GY it reacheD up to 24.4% [153]. It should be noted that even when radionecrosis occurred in about 1/3 of cases, it DID not require treatment because it is asymptomatic and the need for surgical intervention DID not exceed 30% of cases. Finally, combination therapy has been used, including with the use of the FDA-approved bevacizumab for the treatment of recurrent gliomas [154,155]. Interestingly, the combination of stereotactic radiotherapy with bevacizumab also lowers the risk of radionecrosis and reduces the percentage of potential side effects [156,157,158]. It should be noted that the use of SRS, SFRT, or conventional stereotactic radiotherapy with a fractionated doses of 1.8–2.5 GY shows similar results. The choice of re-radiation pattern should be adapted to the volume of the irradiation area and the proximity of sensitive critical organs, taking into account the radiation doses received by them earlier.

### 5.2. Preoperative Radiotherapy

Intensification of postoperative treatment after standard radiochemotherapy by adding SRS with a dose of 15–24 Gy did not contribute to the improvement of the median survival, which was comparable in both groups of the RTOG 9305 study and amounted to approximately 13.5 months [159]. A new trend in the so-called perioperative application of stereotaxic radiotherapy in glioblastoma is related to neoadjuvant therapy. The use of stereotaxic radiotherapy before surgery has several potential benefits that cannot be overestimated for any radiotherapist involved in radiotherapy of brain tumors. First of all, the exact irradiation area is visible, which is related to its easy definition and smaller irradiation volumes. This may allow for a reduction in doses in healthy organs, which is associated with less-intense radiation reactions and widely understood toxicity, e.g., lower risk of radionecrosis. An important factor is also better oxygenation of the area irradiated before surgery, which, by increasing the chance of causing damage to the DNA double helix, may contribute to greater effectiveness of radiotherapy [160]. It should also be noted that in preclinical models, SRS used before surgery has been shown to increase the immune response against neoplasticity by increasing the concentration of circulating cytotoxic T lymphocytes [161]. The above premises contributed to the design of the currently conducted phase I/IIA clinical trial, the so-called NeoGlioma study (NCT05030298), in which, for the standard treatment regimen according to Stupp in one group of patients 14 days before the planned surgery, radiosurgery with a dose of 10 Gy is used. The study is scheduled to be completed in September 2024 [162].

### 5.3. Adaptive/Phase Stereotaxic Radiotherapy

Another path in the development of stereotaxic radiotherapy in the treatment of brain tumors is the so-called staged radiotherapy, which breaks with the current system of classic dose fractionation. It assumes administration of the radiation dose at longer intervals, optimally after obtaining a satisfactory response in the control imaging examination, in order to adapt the dose to a smaller target volume after partial regression of the tumor dimensions. This procedure may allow irradiation of tumors with large initial dimensions and gradually reduce the area of irradiation with the regression achieved. This approach makes it possible to reduce the dose in critical organs and increase the dose in the tumor area. Importantly, this approach allows the classification of patients with larger tumor sizes than during classically fractionated stereotaxic radiotherapy. 

Higuchi et al. analyzed the use of staged stereotaxic radiotherapy in large brain tumors, i.e., >10 cm^3^, in the 3 × 10 Gy regimen administered at two-week intervals. Two weeks after the first treatment, about 90% of the irradiated lesions had decreased. Before the second and third fractions, there was an 18.8% and 39.8% reduction in tumor size, respectively. Local tumor control at one year was 75.9% at 12 months [163]. Lovo et al. presented the results of patients with large secondary brain tumors presenting symptoms. Patients received the first treatment with a dose depending on the initial size of the tumor. Tumors with a size between 2 and 3 cm were prescribed a dose of 18 Gy, those with a size between 3 and 4 cm a dose of 15 Gy, and patients with lesions over 4 cm received a dose of 12 Gy for isodosis of 50%. The second treatment was administered 30 days after the first. During the second procedure, the aim was to administer a total dose of 30 Gy with a median dose of 12 Gy (9–18 Gy) during the second procedure. Median OS was 24 months (3–32 months). One-year local control was 91% [164]. Reports on the use of stepwise radiotherapy in the treatment of primary brain tumors are limited. Romanelli et al. described a case of a pregnant patient with a rapidly progressing brain tumor diagnosed at week 13 with the features of WHO 3 glioma. The patient was given a dose of 14 Gy in one fraction in order to at least temporarily stop the tumor growth and allow the patient to terminate the pregnancy on time. In control-imaging studies, stabilization was achieved. After termination of pregnancy, the patient received chemotherapy based on Avastin, irinotecan, and temodal. At 6 weeks postpartum, MRI showed progression. The patient was qualified for stereotaxic radiotherapy in the 3x8 Gy regimen. A follow-up MRI of the brain performed 33 months after the end of radiotherapy showed complete regression [165]. 

Lovo et al. presented promising results for two-step radiosurgery for large-sized primary brain tumors. The tumors were located in problematic locations for surgeons, e.g., close to the pineal gland or the hypothalamus region. Tumors with dimensions up to 51.6 mL were qualified. In the first stage, a median dose of 13 Gy was used, and after 30–42 days, the patients underwent the second stage of treatment. At the second stage, the tumor size was reduced by an average of 26.4% with an average tumor size of 5 mL (0.1–17.8 mL). The median dose prescribed during the second stage was 16.2 Gy. During the follow-up period, which averaged 336.5 days (65–962 days), 5 out of 8 tumors regressed, 2 out of 8 stabilized, and one patient died after 435 days. No excessive treatment-related toxicity was observed [166]. The optimal dose selection requires further research. To summarize the results obtained so far during ISRS2022, a dose of 2–3 cm in the first stage and 12 Gy in the second stage was proposed for tumors, 12 Gy in the first stage of 3–4 cm tumors, and 18 Gy in the second stage or after 15 Gy, three stages of 10 Gy each in tumors over 4 cm [167].

## 6. Discussion

As statistics show, despite the continuous development of treatment technology, glioblastoma remains a challenge for modern medicine [1,7,10]. The standard management involving surgical resection is highly problematic due to the high malignancy and the tendency of tumor cells to migrate [18,19,20]. For this reason, it is important to develop systemic therapies that support treatment and sometimes represent the only therapeutic option [2,22]. The standard management of primary tumors consisting of radiotherapy combined with temozolomide chemotherapy has repeatedly been shown to contribute to prolonged patient survival [19,24,25]. However, this therapeutic option is associated with a number of limitations regarding the patient’s initial condition, age, and also the genetic profile of the tumor. The treatment of disease recurrence also remains a particular problem. In the case of radiotherapy, the radiation dose adopted during the initial treatment is a key issue. Promising results are being obtained from studies investigating the use of stereotactic radiotherapy in patients with recurrent tumors. However, the selection of an appropriate dose and the timing of treatment still remain a challenge for modern medicine [150,151]. The presence of similar limitations raises the need to develop new, effective treatments options available to a wide range of patients. Molecular targeted therapy and immunotherapy appear to be encouraging approaches.

In recent years, a number of key metabolic points for the progression of GBM have been identified [42,43,44,45,46,47,48]. Control of their activity may result in a reduction in tumor growth, contributing to an improvement in the therapeutic effect of the applied treatment. One of these appears to be the *BRAF* gene, a mutated form of which is mainly characteristic of young GBM patients [47,51]. It encodes a serine/threonine-protein kinase involved in the prooncogenic RAS/RAF/MEK/MAPK signaling cascade. Blocking this signaling through the use of B-raf inhibitors has been shown to improve clinical outcomes for cancer patients. Their use has been approved in the treatment of patients with melanoma, non-small cell lung carcinoma, and thyroid cancer, but recent reports also show their potential use in the therapy of GBM [50,55,56]. One of particular interest to researchers is Vemurafenib, which is a selective *BRAF* V600 inhibitor. The VE-BASKET study showed a positive effect of its use in treatment of gliomas. However, it should be noted that patients with different mutant gliomas of any grade were included in the observation, which makes a conclusive interpretation of the results difficult [58]. Another point of the cascade mentioned above that could potentially be controlled is mitogen-activated extracellular signal-regulated kinase (MEK) [84,85,86]. The observations carried out so far have shown that blocking MEK signaling in GBM causes a decrease in tumor cell proliferation and a decline in percentage of cells positive for Ki67. One of the drugs that is a selective MEK inhibitor is cobimetinib. To date, this compound has been used in the treatment of patients with unresectable or metastatic *BRAF* V600E/V600K-mutated melanoma [128]. The association of MAPK pathway dysregulation in glial tumors also raised hopes for its use in this type of cancer. However, observations from the iMATRIX-cobi trial did not demonstrate the efficacy of cobimetinib monotherapy for patients with HGG. In addition, the researchers were concerned about the high incidence of treatment-related side effects [61]. Despite the unfavorable results relating to the use of MEK inhibitors in monotherapy, an interesting option seems to be the simultaneous use of two compounds affecting tumor metabolism at different points. Another known MEK inhibitor is trametinib. To date, it has been approved in monotherapy for the treatment of melanoma [46,87]. However, research into its use in the treatment of other cancers including GBM is ongoing. The concomitant use of trametinib together with dabrafenib, which is a selective *BRAF* inhibitor, has been shown to improve clinical outcomes (PFS) as well as promote better tolerability by patients and the reduction of resistance development compared with monotherapy [124,125,126]. The efficacy of such a dual-target therapy was also demonstrated for HGG patients in the Rare Oncology Agnostic Research (ROAR) [59]. 

Another interesting concept of controlling tumor metabolism is the influence on angiogenesis, thereby limiting the rate of tumor growth. Vascular endothelial growth factor (VEGF) appears to be crucial in this aspect. As studies have shown, this molecule not only has a direct effect on the rate of progression of GBM, but also influences the development of vasogenic oedema, which is of key importance in the clinical deterioration of patients [109,111]. For this reason, compounds affecting VEGF expression have been of interest to investigators for years. The first, and so far only, VEGF inhibitor approved by the FDA is Bevacizumab as a treatment option in recurrent GBM in adults. It is a humanized, monoclonal anti-body directed against VEGF-A. A phase II study NCI 06-C-0064E conducted in 2009 on patients with recurrent GBM showed a positive effect of bevacizumab on patient survival and recurrence-free survival with a low rate of side effects [71]. Research is currently underway to optimize its effect, including through concomitant administration with other anticancer drugs. One of these is the observations of Wick et al. on the use of Bevacizumab together with lomustine in patients with recurrent GBM. They showed a better clinical response and an improvement in the recurrence-free period. However, it should be noted that there was an increase in the incidence of serious adverse effects compared with taking lomustine as a monotherapy. At the same time, in contrast to the first study by these authors, the effect on the overall survival of patients was not significant [69]. In another study, Friedman et al. analyzed the results of treatment with bevacizumab in combination with irinotecan compared to use in monotherapy in patients with recurrent GBM. They showed a higher PFS rate for patients taking both drugs. However, despite the significant anti-tumor activity of both drugs, the effect on overall survival of patients remained weak. There was even a reduction in OS when irinotecan was included. Additionally, it was associated with an increased incidence of side effects [70]. Interestingly, studies on the use of stereotactic radiotherapy showed longer OS values and lower radionecrosis and adverse effects following the use of bevacizumab [150,156,157,158]. 

These results demonstrate the complexity and multifaceted nature of anticancer therapy. Despite the apparent synergistic actions of the drugs, the therapeutic results noted in patients are often incomplete and, in addition, have a higher risk of adverse effects. Another drug with anti-angiogenic activity is apatinib, which is a selective VEGFR-2 inhibitor. Its anti-tumor effect against glioblastoma cells has been demonstrated both in vitro and in vivo [142]. This was confirmed by the observations of Wang et al. performed in patients with recurrent GBM treated with apatinib and temozolomide. The authors demonstrated the efficacy of treatment evident in the clinical response of patients, improved survival, and extension of the relapse-free period, with a good safety profile of the therapy [78]. Research into other selective VEGF inhibitors is also underway. One of them is Lenvatinib. Preliminary results seem promising, but an accurate analysis of the treatment efficacy effects requires further follow-up in the following years and expansion of the study cohort [77]. Less-selective VEGF inhibitors, such as regorafenib, an inhibitor of multiple kinases involved in tumor angiogenesis, including VEGFR 1–3, have also drawn the attention of researchers. The REGOMA trial showed an encouraging effect of monotherapy with this drug on OS compared with lomustine monotherapy in patients with recurrent GBM [79]. However, it should be noted that the survival of patients taking lomustine in this study differed significantly from observations reported in the past. In other studies also using this drug in the control group, an average survival of 8–10 months was usually achieved [168]. This discrepancy may to some extent distort the authors’ conclusions. Promising initial anti-tumor activity in the treatment of GBM with a good safety profile has also been shown for sunitinib [140].

Not all of the VEGF inhibitors, despite positive results from preclinical studies, have proven their efficacy in the treatment of patients with GBM. An example is afilibercept. In studies conducted on GBM patients, it was not possible to achieve the primary endpoint of the trial, while finding minimal antitumor activity despite the adverse effects of treatment [68]. Better tolerability of the applied treatment was demonstrated in the North American Brain Tumour Consortium Study in the case of pazopanib, which is a second-generation multi-targeted tyrosine kinase inhibitor with activity on several receptors including VEGFR1, VEGFR2, and VEGFR. Unfortunately, these observations also did not demonstrate an effect of pazopanib monotherapy on PFS of treated patients [72]. Unfavorable clinical outcomes have also been obtained from studies on the inclusion of nintedanib in the treatment of recurrent GBM [66]. Similar results were reported with sorafenib, which is also a multitargeted tyrosine kinase inhibitor that, in addition to VEGFR, also acts on PDGFR, and the Ras/Raf signaling pathway. Despite good clinical tolerability, no measurable treatment effect has been found for either patients treated for primary GBM or relapsed disease [73,74]. In the treatment of patients with primary GBM, the use of everolimus, which is a proliferation signal inhibitor in the mTOR pathway, has also been attempted. Unfortunately, despite promising treatment results in other malignancies, everolimus addition has not improved patient survival despite good tolerance of the therapy [63,64]. Primary GBM was also investigated by researchers analyzing the use of paxalisib as an adjuvant treatment after surgical resection and chemoradiotherapy with temozolomide. Patients with newly diagnosed GBM with unmethylated *MGMT* promoter status were observed. In this case, the addition of the inhibitor contributed to prolonged OS and PFS, with good treatment tolerance as well [62]. 

As the above review shows, the results of the application of the analyzed inhibitors are highly variable despite the encouraging conclusions of previous preclinical studies. Glioblastoma is a highly malignant type of tumor characterized by a number of distinct metabolic pathways. At the same time, they are a very heterogeneous group in terms of their molecular profile, which significantly hinders the development of a standard therapy model. Medical developments make it possible to undertake immunohistochemical tests and precisely identify the key metabolic pathways of tumor cells. New drugs offer the chance to target key points in tumor progression. However, the complexity of oncogenetic processes still remains an enigma to modern medicine, making further observations necessary. Studies conducted to date, although indicating the efficacy of these inhibitors in the treatment of patients with HGG, are mostly based on relatively small and, in addition, heterogeneous groups of patients. Due to the high level of complexity of tumor metabolism and the overlap of often several metabolic pathways, more detailed studies on larger groups with a precisely defined molecular profile of the tumors are necessary. 

## 7. Conclusions

Modern targets for targeted drugs and new radiation strategies, along with the development of the possibility of re-radiotherapy, promise to further improve the treatment outcomes of patients with brain gliomas. We need better-designed prospective randomized trials, and the potential of radiotherapy to overcome possible resistance to modern targeted drugs is an interesting prospect.

## Figures and Tables

**Table 1 cancers-14-05377-t001:** Targeted therapies in glioblastoma, currently under clinical trial or completed.

Medication	Target	Population	Phase	Comedication	Overall Survival (OS)	Progression-Free Survival (PFS)
Vemurafenib [58]	V600E *BRAF* V600D *BRAF* V600R *BRAF*	Patients with BRAF-V600-mutant glioma in any point of treatment	Phase II	n.a.	11.9 months (95% CI, 8.3 to 40.1 months) for malignant diffuse glioma	5.3 months (95% CI, 1.8 to 12.9 months) for malignant diffuse glioma
Dabrafenib [59,60]	V600E *BRAF* V600D *BRAF* V600R *BRAF* V600K *BRAF*	Patients with recurrent or progressive *BRAF* V600E–mutant HGG and LGG	Phase II	trametinib	17.6 months (95% CI, 9.5 to 45.2 months) for HGG	3.8 months (95% CI, 1.8 to 9.2 months) for HGG
		Patients with BRAF-V600-mutant solid tumors (including GBM), lymphomas, or multiple myeloma	Phase II	trametinib	28.6 months for all types of cancers	11.4 months (90% CI, 8.4 to 16.3 months) for all types of cancers
Trametinib [59,60]	MEK1/2					
Cobimetinib [61]	MEK1	Paediatric and young adult patients with relapsed or refractory solid tumors (including HGG)	Phase I/II	n.a.	not reached	14.8 months (95 % CI, 3.6 to 14.8) for all types of cancers
Paxalisib [62]	PI3K/mTOR	Patients with newly-diagnosed GBM with unmethylated MGMT promoter status following surgical resection and initial chemoradiation with temozolomide	Phase II	n.a.	15.7 months	8.4 months
Everolimus [63,64,65]	mTOR	Patients with newly diagnosed GBM	Phase II	everolimus + radiotherapy + temozolomide vs. radiotherapy + temozolomide	16.5 months (95% CI, 12.5 to 18.7 months) vs. 21.2 months (95% CI, 16.6 to 29.9 months)	8.2 months (95% CI, 6.5 to 10.6 months) vs. 10.2 months (95% CI, 7.5 to 13.8 months)
				everolimus + radiotherapy + temozolomide	15.8 months (95% CI, 13.0 to 20.3 months)	6.4 months (95% CI, 5.4 to 9.0 months)
		Patients with newly diagnosed GBM, previously not treated	Phase II	radiotherapy + temozolomide + bevacizumab followed by the combination of bevacizumab + everolimus	13.9 months (95% CI, 12.4 to NA months)	11.3 months (95% CI, 9.3 to 13.1 months)
Nintedanib [66]	VEGFR1-R3 and FGFR1–3	First or second recurrence of GBM in patients previously treated with bevacizumab vs. not treated with bevacizumab	Phase II	n.a.	2.6 months (95% CI, 1.0 to 6.9 moths) vs. 6.9 months (95% CI, 3.7 to 8.1 months)	0.9 months (95% CI, 0.7 to 0.9 months) vs. 0.9 months (95% CI, 0.9 to 2.8 months)
Pemigatinib [67]	FGFR1–3	Patients with recurrent GBM or other primary CNS tumors with an activating FGFR1-3 mutation or fusion/rearrangement	Phase II	n.a.	Ongoing	Ongoing
Aflibercept [68]	VEGF-A/B	Patients with recurrent malignant or anaplastic gliomas that did not respond to temozolomide	Phase II	n.a.	9.8 months	3.0 months (95% CI, 2.0 to 4.0 months)
Bevacizumab [69,70,71]	VEGF-A	Patients with GBM with progression after chemoradiation	Phase III	bevacizumab + lomustine vs. lomustine monotherapy	9.1 months (95% CI, 8.1 to 10.1 months) vs. 8.6 months (95% CI, 7.6 to 10.4 months)	4.2 months (95% CI, 3.7 to 4.3 months) vs. 1.5 months (95% CI, 1.5 to 2.5 months)
		Patients with first or second relapse and GBM progression	Phase II	bevacizumab monotherapy vs. bevacizumab + irinotecan	9.2 months (95% CI, 8.2 to 10.7 months) vs. 8.7 months (95% CI, 7.8 to 10.9 months)	4.2 months (95% CI, 2.9 to 5.8 months) vs. 5.6 months (95% CI, 4.4 to 6.2 months)
		Patients with recurrent GBM after chemoradiation	Phase II	n.a.	7.8 months (95% CI, 5.3 to 13.5 months)	4.0 months (95% CI, 3.0 to 6.5 months)
Pazopanib [72]	VEGFR1-R3	Patients with recurrent GBM	Phase II	n.a.	8.6 months (95% CI, 6 to 11.8 months)	3.0 months (95% CI, 2.0 to 3.5 months)
Sorafenib [73,74,75]	VEGFR2-R3	Patients with newly diagnosed GBM, previously not treated	Phase II	radiotherapy + temozolomide followed by the combination of temozolomide + sorafenib	12.0 months (95%CI, 7.2 to 16.0 months)	6.0 months (95% CI, 3.7 to 7.0 months)
		Patients with recurrent GBM	Phase II	temozolomide	10.4 months (95% CI, 6.0 to 13.8 months),	1.6 months (95% CI, 1.0 to 2.9 months)
		Patients with progressive/recurrent GBM	Phase II	erlotinib	5.7 months (95% CI, 4.5 to 7.9 months)	2.5 months (95% CI, 1.8 to 3.7 months)
Sunitinib [76]	VEGFR1-R2	Patients with recurrent GBM	Phase II/III	sunitinib vs. lomustine	Ongoing	Ongoing
Lenvatinib [77]	VEGFR1-R3	Patients with previously treated select solid tumors (including GBM)	Phase II	pembrolizumab	Ongoing	Ongoing
Apatinib [78]	VEGFR2	Patients with recurrent GBM	Phase II	temozolomide	9.0 months (95% CI, 8.2 to 12.2 months)	6.0 months (95% CI, 5.3 to 7.8 months)
Regorafenib [79]	VEGFR1-3	Patients with relapsed GBM	Phase II	regorafenib monotherapy vs. lomustine monotherapy	7.4 months (95% CI, 5.8 to 12.0 months) vs. 5.6 months (95% CI, 4.7 to 7.3 months)	2.0 months (95% CI, 1.9 to 3.6 months) vs. 1.9 months (95% CI, 1.8 to 2.1 months)
Vemurafenib [58]	V600E *BRAF* V600D *BRAF* V600R *BRAF*	Patients with BRAF-V600-mutant glioma in any point of treatment	Phase II	n.a.	11.9 months (95% CI, 8.3 to 40.1 months) for malignant diffuse glioma	5.3 months (95% CI, 1.8 to 12.9 months) for malignant diffuse glioma
Dabrafenib [59,60]	V600E *BRAF* V600D *BRAF* V600R *BRAF* V600K *BRAF*	Patients with recurrent or progressive *BRAF* V600E–mutant HGG and LGG	Phase II	trametinib	17.6 months (95% CI, 9.5 to 45.2 months) for HGG	3.8 months (95% CI, 1.8 to 9.2 months)) for HGG
		Patients with BRAF-V600-mutant solid tumors (including GBM), lymphomas, or multiple myeloma	Phase II	trametinib	28.6 months for all types of cancers	11.4 months (90% CI, 8.4 to 16.3 months) for all types of cancers
Trametinib [59,60]	MEK1/2	JAK WYŻEJ				
Cobimetinib [61]	MEK1	Pediatric and young adult patients with relapsed or refractory solid tumors (including HGG)	Phase I/II	n.a.	not reached	14.8 months (95% CI, 3.6 to 14.8) for all types of cancers
Paxalisib [62]	PI3K/mTOR	Patients with newly diagnosed GBM with unmethylated MGMT promoter status following surgical resection and initial chemoradiation with temozolomide	Phase II	n.a.	15.7 months	8.4 months
Everolimus [63,64,65]	mTOR	Patients with newly diagnosed GBM	Phase II	everolimus + radiotherapy + temozolomide vs. radiotherapy + temozolomide	16.5 months (95% CI, 12.5 to 18.7 months) vs. 21.2 months (95% CI, 16.6 to 29.9 months)	8.2 months (95% CI, 6.5 to 10.6 months) vs. 10.2 months (95% CI, 7.5 to 13.8 months)
				everolimus + radiotherapy + temozolomide	15.8 months (95% CI, 13.0 to 20.3 months)	6.4 months (95% CI, 5.4 to 9.0 months)
		Patients with newly diagnosed GBM, previously not treated	Phase II	radiotherapy + temozolomide + bevacizumab followed by the combination of bevacizumab + everolimus	13.9 months (95% CI, 12.4 to NA months)	11.3 months (95% CI, 9.3 to 13.1 months)
Nintedanib [66]	VEGFR1-R3 and FGFR1–3	First or second recurrence of GBM in patients previously treated with bevacizumab vs. not treated with bevacizumab	Phase II	n.a.	2.6 months (95% CI, 1.0 to 6.9 moths) vs. 6.9 months (95% CI, 3.7 to 8.1 months)	0.9 months (95% CI, 0.7 to 0.9 months) vs. 0.9 months (95% CI, 0.9 to 2.8 months)
Pemigatinib [67]	FGFR1–3	Patients with recurrent GBM or other primary CNS tumors with an activating FGFR1-3 mutation or fusion/rearrangement	Phase II	n.a.	Ongoing	Ongoing
Aflibercept [68]	VEGF-A/B	Patients with recurrent malignant or anaplastic gliomas that did not respond to temozolomide	Phase II	n.a.	9.8 months	3.0 months (95% CI, 2.0 to 4.0 months)
Bevacizumab [69,70,71]	VEGF-A	Patients with GBM with progression after chemoradiation	Phase III	bevacizumab + lomustine vs. lomustine monotherapy	9.1 months (95% CI, 8.1 to 10.1 months) vs. 8.6 months (95% CI, 7.6 to 10.4 months)	4.2 months (95% CI, 3.7 to 4.3 months) vs. 1.5 months (95% CI, 1.5 to 2.5 months)
		Patients with first or second relapse and GBM progression	Phase II	bevacizumab monotherapy vs. bevacizumab + irinotecan	9.2 months (95% CI, 8.2 to 10.7 months) vs. 8.7 months (95% CI, 7.8 to 10.9 months)	4.2 months (95% CI, 2.9 to 5.8 months) vs. 5.6 months (95% CI, 4.4 to 6.2 months)
		Patients with recurrent GBM after chemoradiation	Phase II	n.a.	7.8 months (95% CI, 5.3 to 13.5 months)	4.0 months (95% CI, 3.0 to 6.5 months)
Pazopanib [72]	VEGFR1-R3	Patients with recurrent GBM	Phase II	n.a.	8.6 months (95% CI, 6 to 11.8 months)	3.0 months (95% CI, 2.0 to 3.5 months)
Sorafenib [73,74,75]	VEGFR2-R3	Patients with newly diagnosed GBM, previously not treated	Phase II	radiotherapy + temozolomide followed by the combination of temozolomide + sorafenib	12.0 months (95%CI, 7.2 to 16.0 months)	6.0 months (95% CI, 3.7 to 7.0 months)
		Patients with recurrent GBM	Phase II	temozolomide	10.4 months (95% CI, 6.0 to 13.8 months),	1.6 months (95% CI, 1.0 to 2.9 months)
		Patients with progressive/recurrent GBM	Phase II	erlotinib	5.7 months (95% CI, 4.5 to 7.9 months)	2.5 months (95% CI, 1.8 to 3.7 months)
Sunitinib [76]	VEGFR1-R2	Patients with recurrent GBM	Phase II/III	sunitinib vs. lomustine	Ongoing	Ongoing
Lenvatinib [77]	VEGFR1-R3	Patients with previously treated select solid tumors (including GBM)	Phase II	pembrolizumab	Ongoing	Ongoing
Apatinib [78]	VEGFR2	Patients with recurrent GBM	Phase II	temozolomide	9.0 months (95% CI, 8.2 to 12.2 months)	6.0 months (95% CI, 5.3 to 7.8 months)
Regorafenib [79]	VEGFR1-3	Patients with relapsed GBM	Phase II	regorafenib monotherapy vs. lomustine monotherapy	7.4 months (95% CI, 5.8 to 12.0 months) vs. 5.6 months (95% CI, 4.7 to 7.3 months)	2.0 months (95% CI, 1.9 to 3.6 months) vs. 1.9 months (95% CI, 1.8 to 2.1 months)

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
