# Peer review of "New Directions in the Therapy of Glioblastoma"

_cancers, 2022, doi:10.3390/cancers14215377_

Round 1

Reviewer 1 Report

In this manuscript by Szklener K et al., the authors have comprehensively provided the state-of-the-art information of the various therapies being studied in Glioblastoma (GBM) along with a clinical background of the therapy in different cancer types including FDA approvals. The authors also provide a detailed background on CNS tumors and GBM including the current survival statistics and available treatment regimens.

The information provided in this review article includes depth and is of great value for clinicians in the field providing the current state of therapy in GBM. Considering a satisfactory response to the following major and minor comments, the overall enthusiasm for accepting this manuscript in the journal ‘Cancers’ is high.

1.     The current accepted term for GBM is Glioblastoma. The authors have used the old notation of glioblastoma multiforme. They should change it to the current form.

2.     The abstract is too generic. The authors should broadly specify the categories of therapies discussed in the review, how these have impacted GBM clinical research and about their future implications.

3.     The Introduction gives detailed information on the statistics of CNS tumors and survival values which although useful information (can be discussed under a separate heading) does not provide a reflection on the paper and how over the years the clinical knowledge in GBM treatment has evolved.

4.     The sub sections in section 3 are grouped under the main section 3. called ‘An overview of Glioblastoma which does not maintain a good flow to the readers. Either the authors should change the main heading or group the subsections under a different heading eg: ‘Kinase signaling and its inhibitors in GBM’.

5.     Throughout the review, the authors should introduce smaller paragraphs in their writing instead of providing one single paragraph.

6.     Is table 2 a continuation of table 1, or does it provide a different set of information? If not, the two can be merged. Also, table 2 has not been cited in the text.

7.     In the beginning of section 4, a small paragraph introducing the different drugs, their categories, and the content of the discussion in the following 14 sub-sections can be included.

8.     The authors should cite the recent review on the mechanistic aspects of drug resistance and recurrence in GBM:

Goenka, A., Tiek, D., Song, X., Huang, T., Hu, B., & Cheng, S. Y. (2021). The Many Facets of Therapy Resistance and Tumor Recurrence in Glioblastoma. Cells10(3), 484. https://doi.org/10.3390/cells10030484.

Author Response

Dear Sir or Madam,

The paper has been revised in line with the comments we received. We followed all the remarks, and enhanced the abstract with a synthesis of available therapeutic options along with an indication of the main paths the clinical trials follow. The main text has also been revised to better reflect the evolution of medical knowledge about the options available. Finally, we improved the logical structure of chapters and merged the tables. We would like to express our gratitude to the reviewers for the detailed comments that allowed us to improve the manuscript. We hope that all the changes we made could be considered as sufficient and the article may be eligible for press.

Reviewer 2 Report

The Authors have review very carefully the literature regarding treatment options and prior and ongoing clinical trials on GBM.

Some minor changes:

- please cite in the reference list the WHO Classification 2021 (and not the WHO 2016)

- when the Authors mention the REGOMA trial, it is important to underlie that is a negative trial. In particular, the REGOMA trial reported a median overall survival of 7.4 months in the regorafenib group and 5.6 months in the lomustine group. The control group with the lomustine is the worst control arm as compared with historical control groups with lomustine, and not representative of the real OS following lomustine (see the Editorial of Roger Stupp Stupp R. Drug development for glioma: are we repeating the same mistakes? Lancet Oncol. 2019 Jan;20(1):10-12. doi: 10.1016/S1470-2045(18)30827-1. Epub 2018 Dec 3. PMID: 30522968). Moreover, it is important to underlie in the text that we are awaiting the results of the AGILE trial that will clarify whether regorarefin is or is not significantly active in GBM

Author Response

Dear Sir or Madam,

The paper has been revised in line with the comments we received. We updated the WHO Classification reference and followed other remarks, especially including trials status. We have tried to outline the problems associated with the REGOMA study and also mention ongoing studies relating to the mentioned agents. We would like to express our gratitude to the reviewers for the detailed comments that allowed us to improve the manuscript. We hope that all the changes we made could be considered as sufficient and the article may be eligible for press.

Reviewer 3 Report

The authors provide an extensive overview of GBM in terms of epidemiology, relevant pathway alterations, and specific drugs targeting these pathways. Particularly, the authors cover and discuss those drugs targeting VEGF, FGFR, and the MAPK, PI3K pathways. Relevant clinical studies evaluating these drugs are also extensively described in the review. This is important because currently, gliomas have not benefited from several therapies that work in other cancers. Therefore, awareness of the results and limitations of previous clinical studies in the context of these brain tumors is relevant. Whereas the review provides important information and summarizes previous clinical trials in gliomas and other cancers, there are some points that need to be addressed.

Please find my comments below:

Major comments

-       The review focuses on drugs targeting major kinases in gliomas. It would be important to emphasize the relevance of studying these drugs compared to other therapies such as immunotherapies or chemotherapies.

-       In section 2 (Current treatment guidelines in advanced glioblastoma), it is recommended to discuss the details of tumor-treating fields for glioma therapy. For instance, how much this new therapy extends PFS and OS. Also, what is the role of TTFs in recurrent GBM?

-       The following sentence needs a reference: “In the vast majority of patients with BRAF mutation, GBM is found next to the ventricular system.”

-       Are there any studies incorporating TTFs or immunotherapies in gliomas with the drugs described in the review?

-       Whereas there is a good flow of the narrative, the manuscript would be greatly benefited from an edition by a native English writer as there are several words and phrases that are not written correctly.

Minor comments

-       Multiforme is no longer used to refer to glioblastoma in the newest WHO classification.

-       In line 18, the word “have” is missing.

-       Please replace “analyse” for “analyze”

-       Please replace “that” for “those” in line 306

-       Please write MGMT, IDH, BRAF, and all the gene names in italics

-       Please replace “cancerogenesis” for “carcinogenesis”

-       The sentence stating that: “gene activating mutations appear in 8% of all GBMs” is repeated in two sections of the review.

Author Response

Dear Sir or Madam,

The paper has been revised in line with the comments we received. We clarified the indicated elements of the article, including a wider reference to the effectiveness of TTF. We also improved the language and took into account the minor issues. Finally, we improved the logical structure of chapters and merged the tables. We would like to express our gratitude to the reviewers for the detailed comments that allowed us to improve the manuscript. We hope that all the changes we made could be considered as sufficient and the article may be eligible for press.

Round 2

Reviewer 1 Report

1.     I appreciate the authors attempt to make the abstract more specific and increase its length. However, the abstract is too big now, close to 700 words. Generally, a 300–350 word abstract should be good enough and good for a viewer’s perspective.

2.     I do not think the authors have responded to Comment no. 5 and 7.

Author Response

Dear Sir or Madam,

The paper has been revised in line with the comments we received. We followed all the remarks, and shortened the abstract under 350 words (simultaneously saving a synthesis of available therapeutic options along with an indication of the main paths the clinical trials follow). Moreover, in the beginning of section 4, a small paragraph introducing the different drugs and their categories was included. We would like to express our gratitude to the reviewers for the detailed comments that allowed us to improve the manuscript. We hope that all the changes we made could be considered as sufficient and the article may be eligible for press.

Reviewer 3 Report

The authors have provided satisfying responses to the comments provided in the initial revision.

There are only some gene names that need to be in italics.

Author Response

Dear Sir or Madam,

The paper has been revised in line with the comments we received. We followed all the remarks, and italicized gene names according to the suggestion. We would like to express our gratitude to the reviewers for the detailed comments that allowed us to improve the manuscript. We hope that all the changes we made could be considered as sufficient and the article may be eligible for press.
